# Pre-equilibrium biosensors as an approach towards rapid and continuous molecular measurements

Nicolò Maganzini[1], Ian Thompson [1], Brandon Wilson [2] & Hyongsok Tom Soh [1,2] ✉

Almost all biosensors that use ligand-receptor binding operate under equilibrium conditions. However, at low ligand concentrations, the equilibration with the receptor (e.g., antibodies and aptamers) becomes slow and thus equilibrium-based biosensors are inherently limited in making measurements that are both rapid and sensitive. In this work, we provide a theoretical foundation for a method through which biosensors can quantitatively measure ligand concentration before reaching equilibrium. Rather than only measuring receptor binding at a single time-point, the pre-equilibrium approach leverages the receptor's kinetic response to instantaneously quantify the changing ligand concentration. Importantly, by analyzing the biosensor output in frequency domain, rather than in the time domain, we show the degree to which noise in the biosensor affects the accuracy of the pre-equilibrium approach. Through this analysis, we provide the conditions under which the signal-to-noise ratio of the biosensor can be maximized for a given target concentration range and rate of change. As a model, we apply our theoretical analysis to continuous insulin measurement and show that with a properly selected antibody, the pre-equilibrium approach could make the continuous tracking of physiological insulin fluctuations possible.

Receptor-ligand interactions are a key component of molecular biosensing technologies. Well-chosen molecular receptors confer high specificity, sensitivity, and compatibility with complex biofluids. For sensors designed to quantify low-abundance targets, high-affinity receptors are generally desirable, as they generate large binding signals at low target concentrations. However, bimolecular interactions are affected by an inherent tradeoff between thermodynamics (i.e., sensitivity) and kinetics (i.e., time resolution). With high-affinity receptors and low target concentrations, it takes more time to achieve equilibrium between bound and unbound receptor states[1]. This is because the affinity of a receptor, as quantified by its dissociation constant ($K_D$), is defined by the ratio of its off-rate to on-rate kinetics, $\frac{k_{off}}{k_{on}}$ (Fig. 1a). The on-rate, $k_{on}$, is subject to a fundamental upper limit that is determined by the physical and structural properties of the

receptor and target; values range between $10^6$–$10^7\,s^{-1}\,M^{-1}$ for typical proteins, with a theoretical upper limit of ~$10^8\,s^{-1}\,M^{-1}$ (refs. 2–4). Consequently, differences in affinity are largely due to differences in $k_{off}$, where higher receptor affinity requires a slower off-rate[2]. Because the rate of equilibration is given by $k_{eq} = k_{on}[T] + k_{off}$, this results in long equilibration times. Most molecular biosensors are 'endpoint' sensors, designed to measure target concentrations in a sample of biofluid at a single point in time. In this context, slow equilibration kinetics can be remedied by allowing more time for the sensor to reach equilibrium, and thus are not a concern beyond introducing greater delay in generating a final sensor readout.

More recently, "real-time biosensors"[5–10] that can continuously measure target concentrations have demonstrated promise as a powerful tool for biomedical applications such as drug dosing[11–20],

[1]Department of Electrical Engineering, Stanford University, Stanford, CA 94305, USA. [2]Department of Radiology, Stanford University, Stanford, CA 94305, USA. ✉e-mail: tsoh@stanford.edu

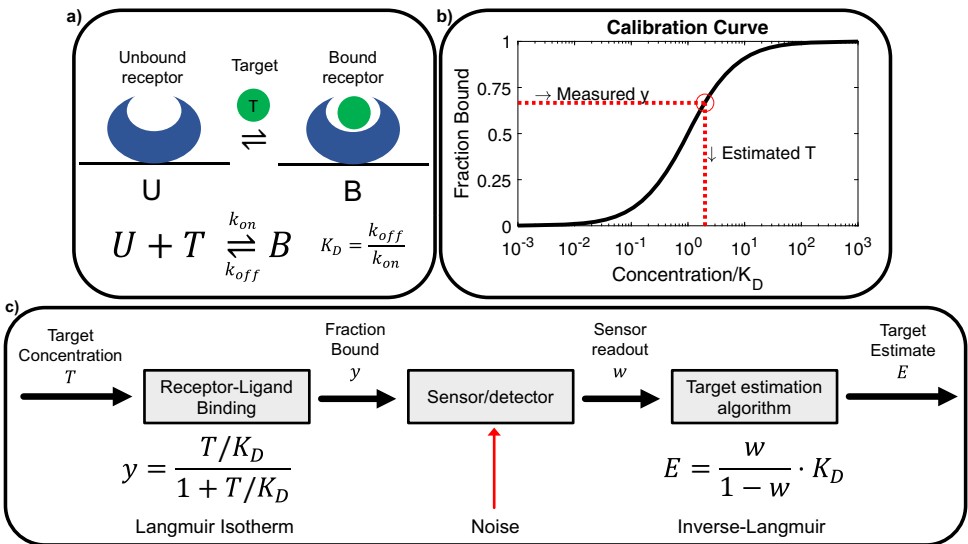

**Fig. 1 | Equilibrium measurements with bimolecular sensors. a** A molecular receptor operating under the bimolecular assumption, where the equilibrium between unbound state U and bound state B is determined by the target concentration $T$. The binding properties of this receptor are described by the kinetic parameters $k_{on}$, $k_{off}$. **b** At equilibrium, the concentration of target ($T$) is back-calculated from the measured signal representing the fraction of target-bound receptor ($y$) based on a calibration curve. **c** A three-component model of this system, wherein $y$ is related to $T$ via the Langmuir isotherm. This $y$ value is measured via the sensor, which also introduces noise. Finally, the target estimation algorithm (TEA) reverses the isotherm to estimate the target concentration.

diagnostics[21–23], and fundamental research[24]. In contrast with endpoint sensors, real-time sensors must adequately balance the tradeoff between thermodynamics and kinetics. Appropriate affinity is required to obtain a meaningful binding signal, but sufficiently fast equilibration kinetics are also critical to achieve reliable quantification of changing target concentrations within biologically relevant timeframes. For this reason, most real-time biosensors developed to date have targeted small-molecule analytes that are present at high concentrations, employing ~µM affinity receptors with fast kinetics, whereas real-time sensing of low-abundance analytes (such as insulin) has remained elusive. Prior efforts to address this problem have focused on increasing the rate of equilibration and tackling mass transport issues. For example, Lubken et al.[1] proposed a novel sensing strategy that leverages small-volume microfluidics to achieve more rapid equilibration. The same authors recently published an insightful analysis of affinity-based real-time sensors that optimizes equilibration kinetics by minimizing diffusion/advection-induced delays between bulk sample concentrations and the surface-bound receptors[25]. However, these approaches assume that equilibration is necessary for accurate quantification of target; here we question whether it is in fact necessary to reach equilibrium binding to achieve real-time analyte quantification.

In this work, we show that it is possible to achieve real-time analyte quantification with biomolecular receptors before reaching equilibrium. Instead of measuring steady-state values, one can observe the equilibration dynamics of the receptor and apply a target estimation algorithm (TEA) to assess analyte concentration. In an ideal, noise-free system, this method could instantaneously determine the target concentration irrespective of the kinetics of the receptor. Noise, however, complicates the measurement of concentration changes that are faster than the kinetics of the molecular system. The sensitivity of pre-equilibrium sensors thus depends on a complex relationship between how rapidly the target concentration is changing and how rapidly the sensor can respond. Here, we investigate the accuracy of noisy biosensors that operate using our proposed pre-equilibrium sensing strategy. We first apply frequency space analysis to examine the way receptors respond to changing target concentrations at different frequencies. We find that such systems act as low-pass frequency filters, wherein slowly changing concentrations (low frequencies) are tracked

better than fast-changing ones (high frequencies), which is in good agreement with prior investigations[25,26] that use similar methods. We then extend this framework to evaluate how the detector noise and the signal attenuation introduced by the molecular receptor impact the ability to reconstruct changing target concentrations. We derive design equations for pre-equilibrium sensor systems and show that the signal-to-noise ratio (SNR) in different sensing conditions can be maximized by an optimum choice of receptor kinetics. As a model, we apply our approach to demonstrate that pre-equilibrium sensing makes it possible to employ antibodies for the task of real-time insulin tracking—an application that poses a daunting challenge for equilibrium-based sensors.

## Results and discussion
### Principles and limitations of equilibrium sensing
Many real-time continuous biosensors apply an equilibrium sensing technique by assuming that the concentration of target, $T$, at any point in time can be estimated from the receptor-bound fraction, $y$, at that time. For a receptor that has reached equilibrium with its surrounding sample environment, $y$ will follow a concentration-dependent equilibrium binding curve determined by the receptor's affinity, $K_D$, which is described by the Langmuir isotherm,

$$y = \frac{\frac{T}{K_D}}{1 + \frac{T}{K_D}}. \tag{1}$$

The target concentration can then be estimated from the measured signal using the inverse of this binding curve (Fig. 1b). Any deviations from this standard curve due to measurement noise will lead to errors in target estimation. Thus, we can describe an equilibrium sensor system as a three-component system: the molecular receptor that binds to the target, the (noisy) sensor instrument that measures the fraction of target-bound affinity reagent, and the target estimation algorithm (here, the inverse-Langmuir standard curve) that back-calculates the target concentration based on this fraction (Fig. 1c).

When this measurement technique is employed, the equilibrium assumption is critical. If measurements are performed before the

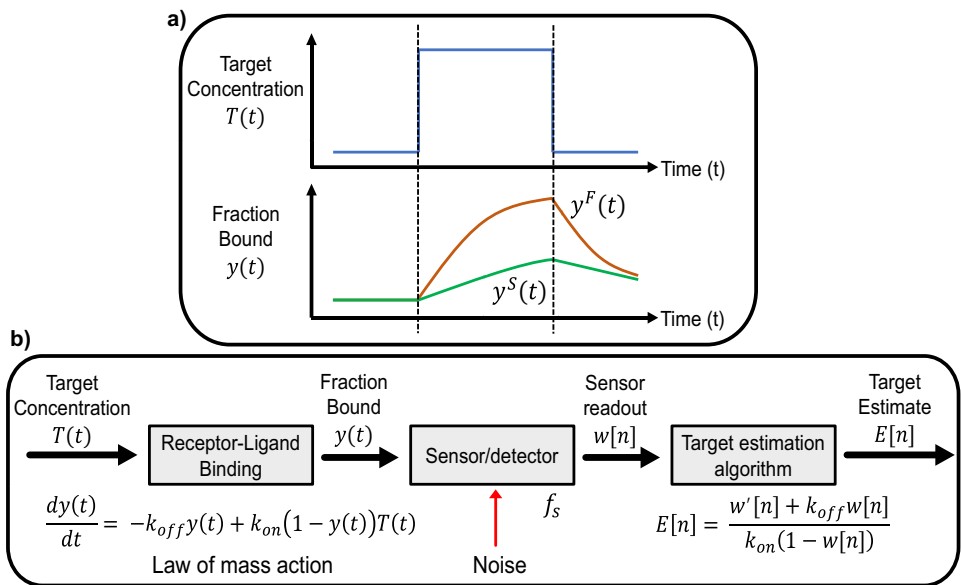

**Fig. 2 | Pre-equilibrium sensing. a** In a short time interval, a molecular receptor with fast kinetics will accumulate more signal, $y^F(t)$, in response to a rapid change in target concentration, $T(t)$, than one with slow binding kinetics, $y^S(t)$. **b** An updated three-component sensor system model that incorporates pre-equilibrium equations and parameters.

receptor has reached equilibrium, the equilibrium binding curve will not accurately describe the assay results, leading to substantial errors. This places a high burden on the kinetics of the molecular receptor. The receptor must reach equilibrium at a sufficiently fast timescale compared to the changing target concentration, such that the sensor's response directly correlates with changing molecular concentrations. The rate of equilibration, $k_{eq} = k_{on}[T] + k_{off}$, is bounded by fundamental on-rate limitations from diffusion and off-rate limitations imposed by the need for a very low $K_D$ to enable sensitive target detection. Thus, there are fundamental limitations to the utility of an equilibrium-based strategy for measuring rapid concentration changes in low-abundance analytes.

**Pre-equilibrium sensing**

By considering the pre-equilibration dynamics of the bimolecular system, we can relax the onerous requirement that the receptor exhibits molecular kinetics that are faster than the rate of change of target concentrations. The law of mass-action describes the time-dependent behavior of a bimolecular receptor through the differential equation:

$$\frac{dy(t)}{dt} = -k_{off}y(t) + k_{on}(1 - y(t))T(t). \quad (2)$$

In this expression, we assume that solution-phase target molecules are in great excess relative to the sensor's receptor, such that the solution target concentration, $T(t)$, is unaffected by the fraction of bound receptors, $y(t)$. We also assume that mass transport effects do not limit our sensor's kinetic response, such that $T(t)$ represents both the bulk and the sensor surface concentration of target. This can be achieved through mindful sensor design and chaotic microfluidic mixing[27–30], and thus is realistic for most real-time sensing applications. Greater detail about designs that satisfy both these assumptions can be found in SI Note 1. By re-arranging Eq. 2, one can determine $T(t)$ at any time-point by measuring both the fraction of bound receptors and the rate-of-change of bound receptors, $\frac{dy(t)}{dt}$, irrespective of how close or far the sensor is from equilibrium:

$$T(t) = \frac{\frac{dy(t)}{dt} + k_{off}y(t)}{k_{on}(1 - y(t))}. \quad (3)$$

Note that when $\frac{dy(t)}{dt} = 0$ and the sensor is at equilibrium, we recover the Langmuir isotherm, Eq. 1. Thus, in an ideal noise-free system, a receptor with any kinetic properties could be used to instantaneously determine the target concentration using the TEA given in Eq. 3, irrespective of how rapidly that concentration is changing. Real-world systems are noisy, however, and this makes it more challenging to measure concentration changes that are much faster than the kinetics of the molecular system. This can be intuitively understood by considering Eq. 2 in the simplified context of two systems, a slow system (S) and a fast system (F), initially unbound with $y(t=0) = 0$. The systems employ receptors with the same affinity, $K_D^F = K_D^S$, but system F's receptor has more rapid kinetics, such that $k_{off}^F = 2k_{off}^S$ and $k_{on}^F = 2k_{on}^S$ (Fig. 2a). When exposed to a step change in concentration, the observed rate of change of the signal in F will be twice that of S. As such, the accumulated binding signal of F in a short interval of time will be twice that of S, even though both systems would produce an equivalent signal at equilibrium. In a noise-free scenario, Eq. 3 would correctly reconstruct $T(t)$ for both systems. In a real system, a fixed amount of noise would impact the slow sensor more due to its lower signal, decreasing the signal-to-noise ratio and increasing the proportion of noise in the estimated target concentration $E(t)$. This implies that for pre-equilibrium measurement systems, sensors with slow kinetics can tolerate less noise than sensors with fast kinetics. In other words, a fast receptor will be able to measure faster-changing target concentrations more accurately than a slow receptor, given the same amount of noise. Thus, while the sensitivity of equilibrium-based sensors is only dependent on noise level and $K_D$, the sensitivity of pre-equilibrium sensors will depend on a complex relationship between how rapidly the target concentration is changing and how rapidly the sensor can respond. This relation will in turn influence how noise propagates through the TEA, and due to the non-linearity of the Langmuir isotherm underlying receptor binding, these relationships will also depend on the thermodynamics of the system. Understanding and quantifying these relationships is essential to the successful realization and optimization of pre-equilibrium-based sensors.

Based on this understanding, we updated our initial three-component sensor system model with the pertinent pre-equilibrium sensing equations (Fig. 2b), where the target concentration $T$ and receptor-bound fraction $y$ are allowed to be continuous variables in

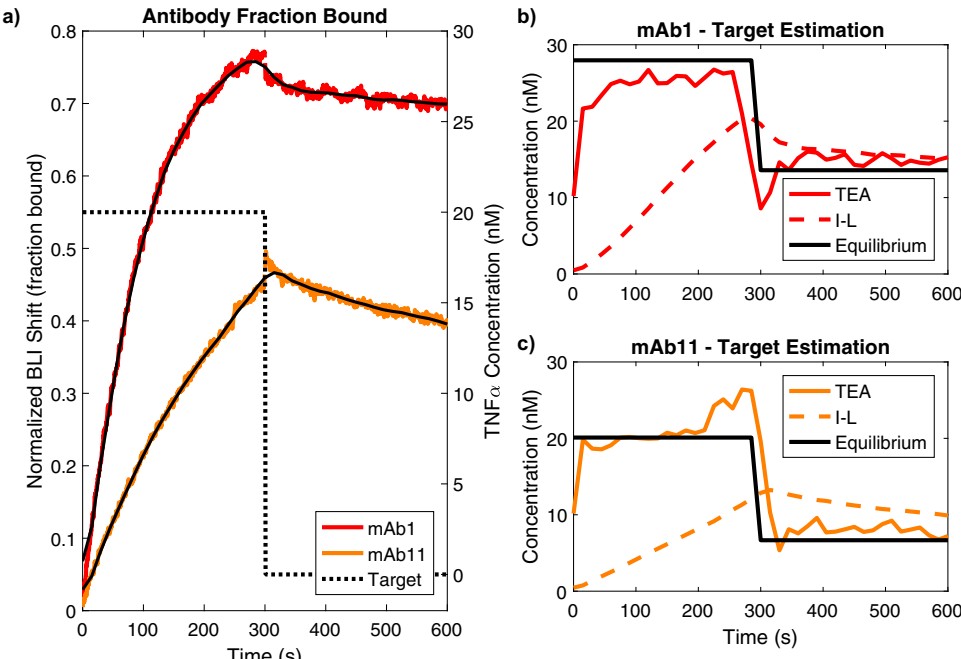

**Fig. 3 | Experimental proof of principle. a** Normalized biolayer interferometry (BLI) binding data for two monoclonal antibodies, mAb1 (red) and mAb11 (orange), exposed to 20 nM TNFα from 0–300 s and 0 nM TNFα from 300–600 s. Black lines represent digitally low-pass filtered data. The target concentration was estimated using the pre-equilibrium TEA (solid lines) as well as the conventional inverse-Langmuir (dashed lines) for **b** mAb1 and **c** mAb11 data. Solid black lines represent the inverse-Langmuir target estimates based on the fraction of bound receptors at equilibrium.

time. In this model, the sensor samples these continuous-time signals with sampling frequency $f_s$ and generates a discrete-time sequence of measurements, $y[n]$, where the integer variable $n$ replaces the continuous variable $t$. Measurement noise, $N[n]$, is an additive term introduced by the sensor, leading to the noisy sensor output $w[n] = y[n] + N[n]$. The revised TEA is the re-arranged law of mass action (Eq. 3) applied to $w[n]$, which generates a discrete-time sequence of target estimates, $E[n]$. In the absence of noise, this system would accurately track the target concentration regardless of receptor kinetics – we used this model to understand how measurement noise impairs accurate target estimation. Furthermore, later in this work we show that, while the pre-equilibrium method could be applied to sensors functionalized with any receptor, measurement precision can be maximized by optimizing the kinetic properties of the receptor employed by the sensor.

Importantly, throughout this work, we assume that the sensor readout $w[n]$ is in units of fraction bound, with values in the range of 0 to 1. To achieve this, a real-world sensor system requires calibration of the raw detector readout (in units of current, voltage, fluorescence intensity, etc.) to determine its correlation with fraction bound. This often involves normalization and background subtraction, and inaccuracies in this process may introduce systematic errors in the readout, which will increase the error in the estimation of target concentrations. Systematic errors may also be introduced by phenomena such as sensor drift and offsets due to non-specific binding. As will be shown later, these sources of inaccuracy affect pre-equilibrium and equilibrium sensors equally due to their low-frequency nature. Therefore, this work will instead focus on analyzing the impact of random noise introduced during measurement, which impacts the proposed pre-equilibrium estimation method more significantly than it does conventional inverse-Langmuir estimation.

### Validation of pre-equilibrium sensing through a biolayer interferometry experiment

To validate the working principle of pre-equilibrium measurement, we carried out a biolayer interferometry (BLI) experiment. We measured

the binding signals of two monoclonal antibodies—mAb1 ($k_{on} = 3.80 \pm 0.07 \times 10^5 \, s^{-1} \, M^{-1}$, $k_{off} = 2.5 \pm 0.3 \times 10^{-3} \, s^{-1}$) and mAb11 ($k_{on} = 1.39 \pm 0.05 \times 10^5 \, s^{-1} \, M^{-1}$, $k_{off} = 2.1 \pm 0.5 \times 10^{-3} \, s^{-1}$) – that were exposed to a step in target concentration: 20 nM followed by 0 nM TNFα (Fig. 3a). We then applied pre-equilibrium estimation as well as the inverse-Langmuir to estimate the target concentration at every instant in time (Fig. 3b, c). We found that, although the BLI sensor did not fully reset when exposed to the blank solution due to non-specific binding, the proposed pre-equilibrium method achieved much more rapid target quantification compared to the inverse-Langmuir.

During this experiment, the sensor did not reach equilibrium due to insufficiently rapid binding kinetics. However, for purposes of comparison, the raw sensor data was fitted to association and dissociation exponential curves using the BLI analysis software and extrapolated to determine the fraction bound at equilibrium (see SI Note 2). We used these values to determine the hypothetical inverse-Langmuir target estimate, had the receptor kinetics been fast enough to reach equilibrium. These estimates are shown in Fig. 3b, c as solid black lines. We note that there is some discrepancy between the true target concentrations used and the equilibrium estimates due to non-specific binding, particularly in the second step, where the sensor does not equilibrate back to 0 fraction bound.

This data demonstrates how the pre-equilibrium analysis can reach the same estimate as the inverse-Langmuir without the need of equilibration. As would be done in a real biosensor, the noisy sensor data was digitally low-pass filtered to remove as much noise as possible while preserving the signal shape (Fig. 3a, black lines). First, the inverse-Langmuir was applied to this filtered data to show that, because the receptor's kinetics are slower than the target concentration changes, the instantaneous concentration estimated by the inverse-Langmuir (Fig. 3b, c, dotted lines) is highly inaccurate. Conversely, using the same sensor data, the proposed pre-equilibrium method, which leverages Eq. 3, instantaneously estimated a target concentration that was much closer to the equilibrium estimate, for both the faster and slower receptors (Fig. 3b, c, solid lines). More proof-of-concept data is included in SI Note 2. However, it should be

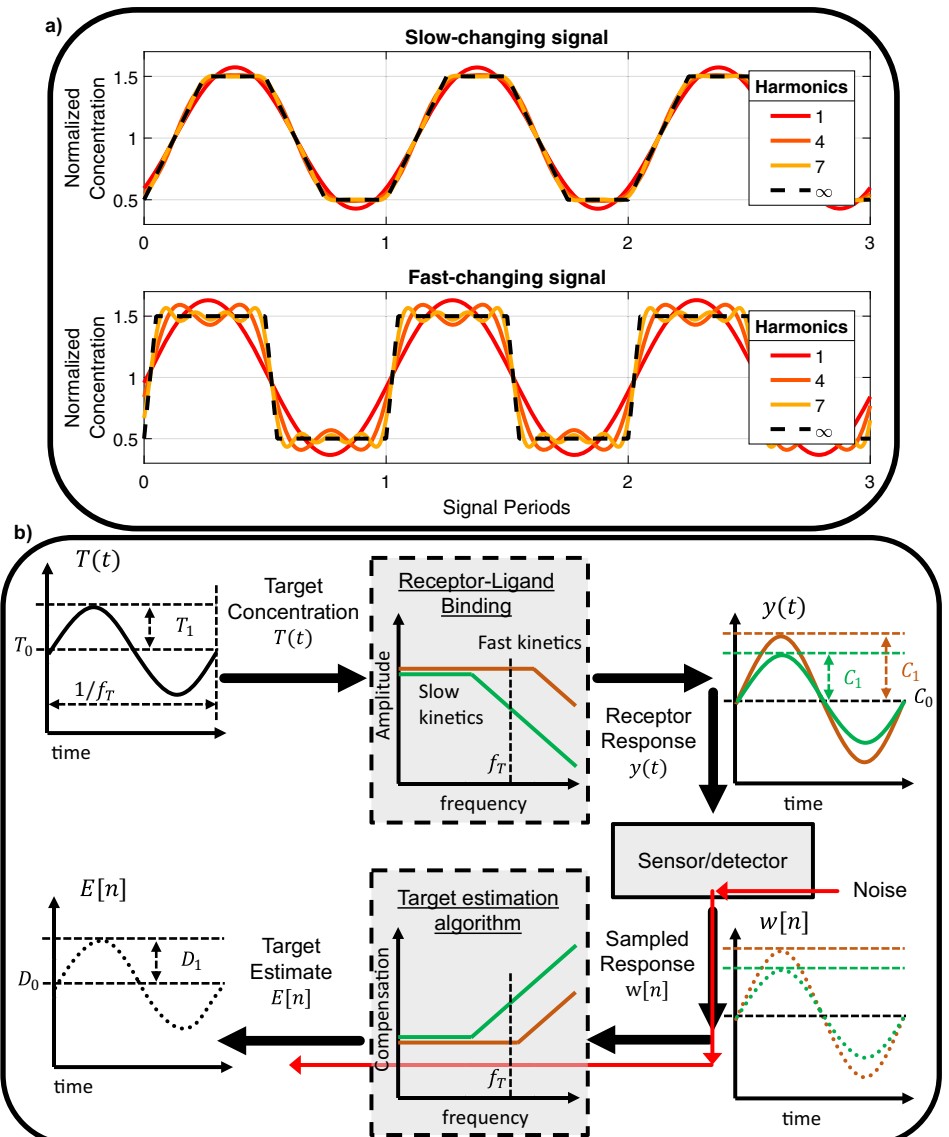

**Fig. 4 | Frequency-domain analysis of a pre-equilibrium sensing system. a** Slow-and fast-changing target concentration signals resolved with increasingly high-frequency content. High-frequency content is associated with fast-changing features in the waveform. A larger number of harmonics are required to resolve the sharp transitions of the fast-changing signal. **b** We used a sinusoidal test signal $T(t)$ with mean concentration $T_0$ and amplitude $T_1$ to analyze the receptor's frequency-domain characteristics. A receptor with fast kinetics will respond equally well to low- and high-frequency content. A receptor with slower kinetics will attenuate high-frequency content, however, resulting in a reduced fraction of target-bound receptors, $y(t)$. To reconstruct the original target signal $T(t)$ from the sampled response $w[n]$, the TEA must compensate for this attenuation. Since noise introduced by the sensor will experience this amplification as well, compensation of slow-kinetics systems will also increase the contribution of noise to the target estimate.

noted that the pre-equilibrium estimates display noticeable variance about their mean, which reduces the precision of the measurement. As introduced in the previous section, the precision of pre-equilibrium measurements depends on the amount of noise in the detector relative to the rapidity of the receptor's kinetics. Conversely, low-frequency offsets such as the non-specific binding shown in this experiment impact pre-equilibrium and inverse-Langmuir estimates equally. In the analysis that follows we explore and characterize the frequency dependence of pre-equilibrium estimation in the presence of noise to inform the design of sensor systems that leverage this technique.

**Analysis of the pre-equilibrium sensing system is based on a frequency-domain approach**

To assess our pre-equilibrium sensor model, we employed frequency-domain analysis[31]. This approach is built upon Fourier analysis, which allows us to approximate any target waveform using a sum of sinusoids of different frequencies. As we increase the number and frequency of the harmonics used in the approximation, we can capture rapidly changing features of the target waveform more accurately. Figure 4a shows two signals—one with slow transitions and one with sharp concentration changes—overlayed with their corresponding approximations using increasing numbers of frequency harmonics. A higher number of harmonics was required to resolve the sharper edges of the fast-changing signal. Lubken et al.[25] provide an analysis method that reconciles physiological concentration change rates (CCR) with corresponding frequency content for a variety of high-profile targets. By analyzing the frequency response of a receptor, we can quantify the extent to which the receptor will resolve rapid changes in the target concentration waveform. Intuitively, we expect that a receptor with fast kinetics, which rapidly reaches equilibrium, will respond to high and low frequencies in a similar manner. In contrast, the response of a receptor with slow kinetics will be impaired or attenuated at high

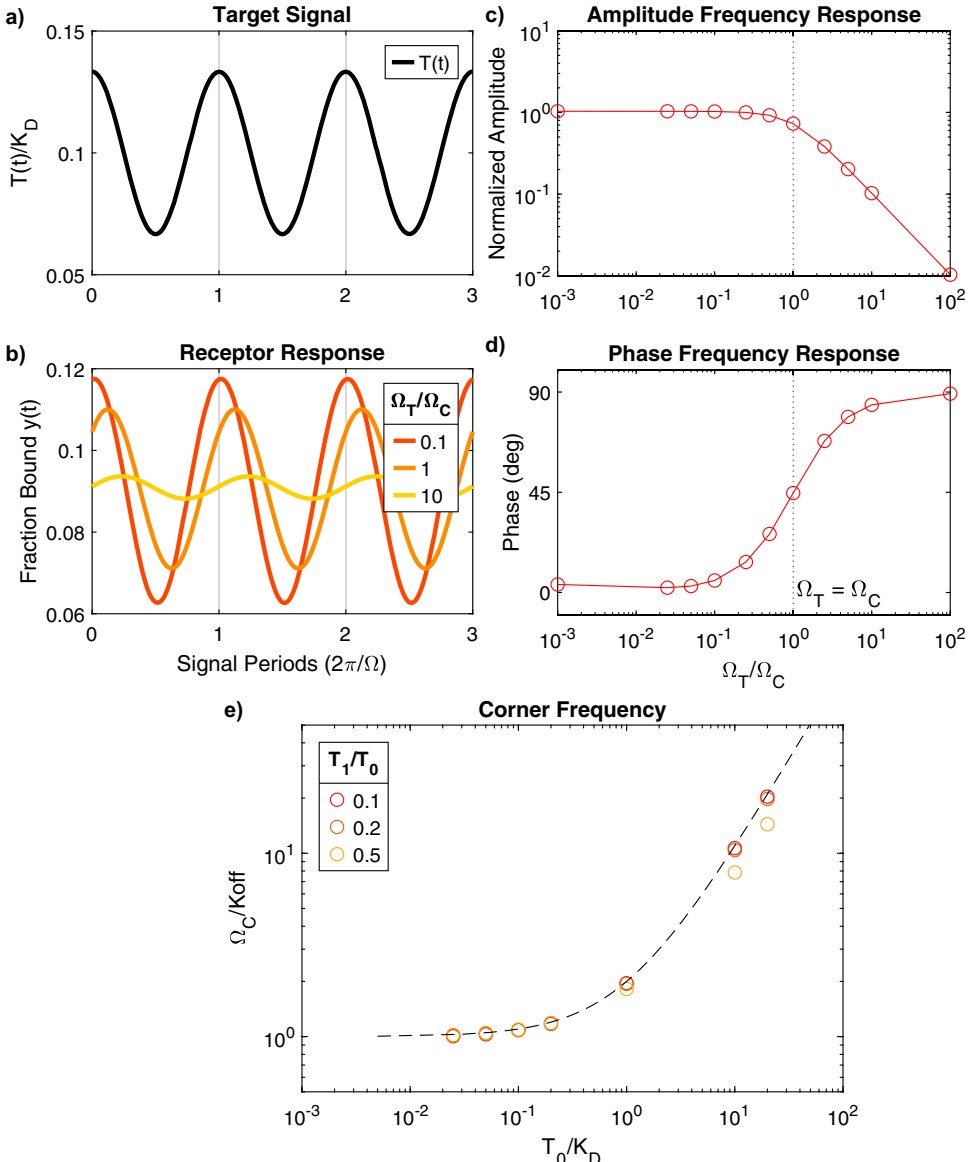

**Fig. 5 | Numerical simulations of a bimolecular sensor. a** Example target signal employed in a simulation with $T_0/K_D = 0.1$ and $T_1/T_0 = 1/3$. Target waveforms were used as inputs to Eq. 2 to numerically simulate the response of the receptor. **b** Simulated steady-state responses of the receptor to the target signal in (**a**). Waveforms represent three different systems with progressively slower kinetics, where the ratio of target frequency ($\Omega_T$) to corner frequency ($\Omega_C$) increases from 0.1 (fastest kinetics) to 10 (slowest kinetics). **c** Amplitude and **d** phase measured from numerical simulations of a range of systems with progressively slower kinetics (increasing $\Omega_T/\Omega_C$). Amplitude is normalized to the expected amplitude of a system at equilibrium. Phase is measured with respect to the target waveform. The frequency response is in good agreement with a first-order low-pass filter model with corner frequency aligned at $\Omega_T/\Omega_C = 1$. **e** Measurements of corner frequency for systems with different TOP. The predicted dependence of $\Omega_C/k_{off}$ on system thermodynamics (Eq. 6) was verified numerically. The dotted line represents Eq. 6, circles are measurements of $\Omega_C$ (normalized to $k_{off}$) from simulated systems with different $T_0/K_D$. Colors indicate target signal amplitude relative to average target concentration ($T_1/T_0$).

frequencies compared to low frequencies, and thus won't resolve the rapidly changing features of the target waveform.

Through knowledge of the receptor's kinetic parameters, one can use the TEA to estimate target concentrations by compensating for the receptor's attenuation of high-frequency components, such that all frequency components of the target estimate are equal to those of the original target signal. However, this compensation will be applied to noise introduced by the sensor as well. For a receptor with slow kinetics, the algorithm must compensate for large signal attenuations, and the noise will be greatly amplified. For a fast receptor, the required amplification of high frequencies is minimal, and thus the impact of noise on the target estimate will be smaller. Thus, the limits on the amount of noise that can be tolerated in a system employing a receptor with slow kinetics will be more stringent than in a system employing a fast receptor. This means that the fast system can resolve higher frequency content than the slow system with the same amount of noise.

To carry out our frequency-domain analysis, we defined a 'test' target signal $T(t)$ that comprises a sinusoidally-varying concentration of amplitude $T_1$ summed to an average concentration, $T_0$:

$$T(t) = T_1 \cos \Omega_T t + T_0, \text{ (in units of molar)}$$

where $\Omega_T = 2\pi f_T$ is the frequency of the sinusoid in rad/s, and $f_T$ is the frequency in Hz (Fig. 4b). We then leveraged an analysis technique termed harmonic balance to obtain mathematical relationships between $T_0, T_1$ and $C_0, C_1$ and $D_0, D_1$, where the $C$ and $D$ parameters

respectively refer to the fraction-bound and target estimate signals, as follows:

$$y(t) = C_1 \cos \Omega_T t + C_0, \text{ (fraction bound, unitless)}$$

$$E[n] = D_1 \cos \Omega_{T,S} n + D_0, \text{ (in units of molar)}$$

where $\Omega_{T,S} = 2\pi f_T / f_S$ is the frequency of the sinusoid normalized to the sampling frequency of the detector. For example, the ratio $C_1/T_1$ quantifies the attenuation of the sinusoidal component introduced by the molecular receptor, while $D_1/C_1$ describes the compensation of the sinusoidal component carried out by the TEA. This analysis assumes that the time-domain response of the system to the periodic sinusoidally-varying input is itself periodic and can be expressed as a Fourier series. In addition, we assume that the magnitude of the oscillation of the fraction bound is small compared to the mean value. We then run simulations to evaluate the accuracy of the model when this is not the case.

## Frequency response of the receptor–slow kinetics act as low-pass filters

We analyze the bimolecular system in the frequency domain by applying the harmonic balance method to Eq. 1. Details of this analysis are reported in SI Note 3. The analysis results in the following relationships between $T_0, T_1$ and $C_0, C_1$:

$$C_0 = \frac{k_{on} T_0}{k_{off} + k_{on} T_0}, \tag{4}$$

$$\frac{C_1}{T_1} = \frac{k_{on}}{k_{off} + k_{on} T_0} \frac{1}{1 + i \frac{\Omega_T}{k_{off} + k_{on} T_0}}. \tag{5}$$

Equation 4, which relates the zero-frequency (constant) term of $T(t)$ to the constant term of the fraction bound signal $y(t)$, is simply the Langmuir isotherm. This confirms that the harmonic balance analysis correctly predicts the behavior of a receptor exposed to a constant target concentration. Equation 5 details the impact of the receptor's kinetics on its ability to respond to target oscillations, in terms of the fraction of target-bound receptors. This equation takes the form of a first-order low-pass frequency filter with a pre-factor and describes the attenuation of the sensor response as the frequency of the target increases. The single pole present in the term has a frequency $\Omega_c = k_{off} + k_{on} T_0$. At this frequency, the second term becomes $\frac{1}{1+i}$, and therefore reduces the amplitude by −3dB (-30%). This frequency is also known as the 'corner frequency' because at lower frequencies, $\Omega_T < \Omega_C$, the magnitude is largely unaffected, but for $\Omega_T > \Omega_C$ the sensor response is subject to steep attenuation (−20 dB/10-fold attenuation per 10-fold increase in frequency). We can parametrize the corner frequency as a system-dependent ratio,

$$\frac{\Omega_C}{k_{off}} = 1 + \frac{T_0}{K_D}. \tag{6}$$

When the average concentration of target is large compared to the affinity of the sensor, the sensor operates at a high $\Omega_C/k_{off}$ and is capable of measuring fast-changing signals (relative to its off-rate) with minimal attenuation. This is defined as a high 'thermodynamic operating point' (TOP). However, when $T_0 \ll K_D$—i.e., at a low TOP—we find that $\Omega_C \approx k_{off}$. Thus, a fast off-rate will be required to observe rapidly changing signals with minimal attenuation.

We performed numerical simulations of a bimolecular sensor using MATLAB[32] to verify agreement between the frequency-domain equations proposed above and the time-domain differential equations that describe the system (Eq. 2). We synthesized test target signals (Fig. 5a) and numerically observed the steady state response of Eq. 2,

which represents the response of an idealized receptor. We simulated three receptors with equal affinity but progressively slower kinetics, and their response is shown in Fig. 5b. All three receptors oscillated about a mean fraction bound of $C_0 \approx 0.091$, which is consistent with the Langmuir relation (Eqs. 1, 4) for $T_0/K_D = 0.1$. The system operating below its corner frequency ($\Omega_T/\Omega_C = 0.1$, red waveform) had the highest-amplitude response to target oscillation, while receptors with slower kinetics showed progressively smaller responses. Receptors with slower kinetics also exhibited a delayed binding response that lags changes in the target concentration, reflective of a shift in the phase of the sinusoid. Both the amplitude and phase behavior are in qualitative agreement with the analytical expression provided above (Eq. 5). We studied this further by simulating a range of systems with different kinetics and measuring their amplitude and phase response (Fig. 5c, d).

When target concentrations changed much more slowly than receptor kinetics ($\Omega_T/\Omega_C \ll 1$), the measured amplitude response was very close to that expected from a bimolecular sensor operating at equilibrium. When the frequency of the target oscillations matched the corner frequency, the amplitude decreased by about 30%, and the phase increased to 45°. As target oscillation frequency increased beyond that, we saw a steep 10-fold decrease in amplitude per 10X increase in frequency. This indicates that the first-order low-pass filter model proposed above, with $\Omega_C$ as its corner frequency, is a good approximation of the behavior of the receptor, as was previously reported[25,26]. Next, we verified that the proposed relation between $\Omega_C/k_{off}$ and the sensor TOP (i.e., Eq. 6). We simulated systems with different TOPs and measured their corresponding $\Omega_C$. Figure 5e shows these measurements, normalized to $k_{off}$, together with a plot of Eq. 6. We also simulated different amplitudes of target oscillation, $T_1/T_0$. For small oscillations, the proposed analytical expression was in good agreement with the simulations. For increasing $T_1/T_0$, however, the system deviated from the assumptions of the analytical derivation and the accuracy decreased, particularly when $T_0 \gg K_D$ and the sensor approached saturation.

The insights and equations provided in this section can be used as heuristics to evaluate the suitability of a given receptor's performance or to estimate the kinetic requirements of a receptor, in the context of particular sensing applications. The challenge of real-time monitoring of insulin offers an interesting case study. Suppose a sensor is intended to track fluctuations in insulin levels, with an average concentration of 100 pM, and that we intend to use a receptor with a $K_D$ of 500 pM. If $k_{on} \approx 10^6$ s$^{-1}$ M$^{-1}$ which is typical of a "good" antibody, this receptor will have a $k_{off} \approx 5 \times 10^{-4}$ s$^{-1}$. Leveraging Eq. 6, we find that $\Omega_C \approx 6 \times 10^{-4}$ rad/s, which corresponds to a period of about 3 h, indicating that signals with periodic behavior faster than 3 h would experience significant attenuation by the receptor. For example, rapid changes in insulin concentration on the scale of 10 min, such as after a meal, would result in the sensor operating well above its corner frequency ($\Omega_T/\Omega_C \approx 20$). This implies that changes occurring on the scale of 10 min would be attenuated in magnitude by approximately 20-fold compared to the response predicted by the Langmuir isotherm. Slower changes, on the scale of 1h, would correspond to $\Omega_T/\Omega_C \approx 3$, or an attenuation of approximately 3-fold compared to the equilibrium response. Although this poses a seemingly intractable problem in the presence of measurement noise, as described below, receptors of this type can still potentially be of value in this context.

## Frequency response of the TEA–slower kinetics result in more noise

We have observed how a receptor affects the frequency content of $T(t)$ in the process of transducing it into $y(t)$. Next, the sensor measures the continuous-time signal from $y(t)$, sampling it at frequency $f_S$ to produce a discrete-time sequence $w[n]$ with values ranging from 0 to 1. The sensor also injects noise, $N[n]$, into $w[n]$. The dominant

components of this noise depend on the nature of the detection modality, but we assume here that $N[n]$ encompasses all relevant sources of random uncertainty. We also make the simplifying assumption that $N[n]$ is white gaussian noise with equal power (given by $N_0/2$ with units of (fraction bound)$^2$/Hz) at all frequencies, and that this noise is independent of the receptor used. White noise is zero-mean and has power spectral density equal to $N_0/2$ at all frequencies $<|\frac{f_S}{2}|$. Because electronics/detectors can operate at very rapid time-scales compared to physiological changes ($f_S \gg 2f_T$), digital filtering or averaging techniques are typically employed to eliminate noise at frequencies higher than $f_T$[33]. At reasonable concentrations for real-time sensing (i.e., pM to nM), there is a relatively large number of bound receptors on the sensor surface, and Poisson noise is assumed to be negligible compared to detector noise. For more information on this assumption, see SI Note 1, where the magnitude of Poisson noise is quantified for an exemplary sensor and shown to be negligible compared to the white noise assumptions made in our later analysis. We evaluated the impact of $N[n]$ on our ability to resolve changing molecular concentrations with slow-kinetics receptors. As shown in Fig. 4, the frequency content of $N[n]$ is shaped by the TEA. Since the receptor attenuates high frequencies in the signal, the TEA applies compensation by amplifying them—the slower the receptor or the higher the frequency, the greater the amplification. $N[n]$ experiences this high-frequency amplification as well.

Thus, to accurately quantify how noise will impair target signal reconstruction, we must analyze the frequency behavior of the TEA. We applied the harmonic balance method to the equation,

$$E[n] = \frac{w'[n] + k_{off} w[n]}{k_{on}(1 - w[n])} . \tag{7}$$

Details are reported in SI Note 4. The analysis results in the following relationships between $C_0, C_1$ and $D_0, D_1$:

$$D_0 = K_D \cdot \frac{C_0}{1 - C_0} , \tag{8}$$

$$\frac{D_1}{C_1} = K_D \cdot \frac{1}{1 - C_0} \left( 1 + \frac{f_S}{k_{off}} \left( 1 - e^{-i\frac{2\pi f_T}{f_S}} \right) \right) . \tag{9}$$

Equation 8, which relates the zero-frequency term of $y(t)$ to the zero-frequency term of the target estimate $E[n]$, is simply the inverse-Langmuir. Equation 9 describes how the TEA reconstructs the target concentration by compensating attenuated frequencies in the signal generated by the bound-receptor fraction. The equation takes the form of a discrete-time high-pass filter[33]. Considering first the limiting case of very low-frequency oscillations, $f_T \rightarrow 0$ (i.e., when the sensor perfectly tracks the target oscillations), $D_1/C_1$ reduces to the frequency-independent pre-factor:

$$\left. \frac{D_1}{C_1} \right|_{f_T = 0} = K_D \cdot \frac{1}{1 - C_0} , \tag{10}$$

which approximates the inverse-Langmuir if $C_1 \ll C_0$. This shows that for slow-changing signals for which the receptor reaches equilibrium, the TEA estimates target concentration simply by applying the inverse-Langmuir without further compensation. At the maximum measurable target frequency ($f_T = f_S/2$),

$$\left. \frac{D_1}{C_1} \right|_{f_T = \frac{f_S}{2}} = K_D \cdot \frac{1}{1 - C_0} \left( 1 + \frac{2f_S}{k_{off}} \right) . \tag{11}$$

Thus, at higher frequencies, the TEA applies an extra compensation term to the inverse-Langmuir that depends on the ratio of $f_S/k_{off}$. Slow receptors with small $k_{off}$ will require more compensation, and therefore noise will be more amplified in such systems.

We performed simulations of the TEA to verify the agreement between the frequency-space equations proposed above and the discrete-time nonlinear differential equation that describes the TEA. We synthesized test fraction-bound signals to model the receptor response (Fig. 6a), sampled the signals, processed them with the TEA (Eq. 7) using $K_D = 500$ pM and $k_{on} = 10^6$ s$^{-1}$ M$^{-1}$ as receptor parameters based on the above insulin-sensing example, and then repeated this for both a slow- and a fast-changing signal (Fig. 6b). As expected, the amplitude of the estimated target signal was greater for the rapidly-changing signal (100–150 pM) compared to the lower-frequency signal (110–140 pM). This is because for a bimolecular sensor to generate equal fraction-bound signal changes at two different target oscillation frequencies, the change in target concentration would need to be larger in the higher-frequency system.

We then simulated four systems with $K_D = 500$ pM and $k_{on}$ of $10^{5.5}, 10^6, 10^{6.5},$ or $10^7$ s$^{-1}$ M$^{-1}$ (and thus varying $\Omega_C$) and measured the amplitude of the target estimates generated by the TEA for a range of $f_T$ values. Figure 6c shows this amplitude after normalization to the inverse-Langmuir equation; values >1 indicate that the TEA is compensating for attenuation introduced by the receptor. For slow-changing target concentrations, where $1/f_T > 2\pi/\Omega_C$, the TEA output was well-aligned with the inverse-Langmuir (i.e., normalized values close to 1). When the frequency exceeded $\Omega_C$, however, TEA compensation increases. Equation 9 proved to be a good model for the frequency response of the TEA, as it accurately accounts for the relationship between $f_T$, $f_S$, and $k_{off}$.

We next leveraged the frequency response of the proposed TEA to obtain a closed-form expression of the amount of noise at the output of the TEA and show its dependence on receptor kinetics and $f_T$. The average noise power at the output of the TEA, $\overline{\sigma^2_{N,Out}}$, is obtained by integrating the noise power at all frequencies (details in SI Note 5). The root mean square (RMS) noise in the target estimate $E[n]$, which represents the average error in the target estimate, is directly related to the average noise power through $RMS_N = \sqrt{\overline{\sigma^2_{N,Out}}}$, in units of moles$_{RMS}$. To isolate the impact of the TEA on the amount of noise observed in $E[n]$, we express the average noise power as:

$$\overline{\sigma^2_{N,Out}} = \frac{N_0}{2} \cdot S_N(f_T, f_S), \tag{12}$$

where the noise scaling function $S_N(f_T, f_S)$, with units of Hz$\left(\frac{moles}{fraction\ bound}\right)^2$, summarizes the amount of compensation applied to the white noise $N[n]$ by the TEA. $S_N(f_T, f_S)$ is given by

$$S_N(f_T, f_S) = (K_D + T_0)^2 \left( f_T \left( \frac{2f_S^2}{k_{off}^2} + \frac{2f_S}{k_{off}} + 1 \right) - \frac{f_S^2}{\pi} \left( \frac{f_S}{k_{off}^2} + \frac{1}{k_{off}} \right) \sin\left( \frac{2\pi f_T}{f_S} \right) \right). \tag{13}$$

More details are provided in SI Note 5. Equation 13 allows us to evaluate important system design decisions directly in terms of the SNR of the output target estimate $E[n]$,

$$SNR_E = \frac{var(T(t))}{\frac{N_0}{2} \cdot S_N} . \tag{14}$$

Importantly, Eq. 14 states that to improve the signal-to-noise ratio of our measurement, we must either reduce noise in our detector (lower $N_0/2$) or reduce the noise scaling function $S_N$ by changing the sensor's parameters.

We evaluated the analysis above by again simulating a sensor for insulin tracking. We generated a trapezoidal target signal that models the rise and fall of insulin concentrations fluctuating between 50 and 150 pM over a period of 10 min (a fundamental frequency of 1/600 Hz) (Fig. 7a). We simulated the response of three receptors with $K_D = 500$ pM but progressively slower kinetics (Fig. 7b). We then sampled $y(t)$ and added noise $N[n]$ with moderate standard deviation ($\sigma_N = 0.005$

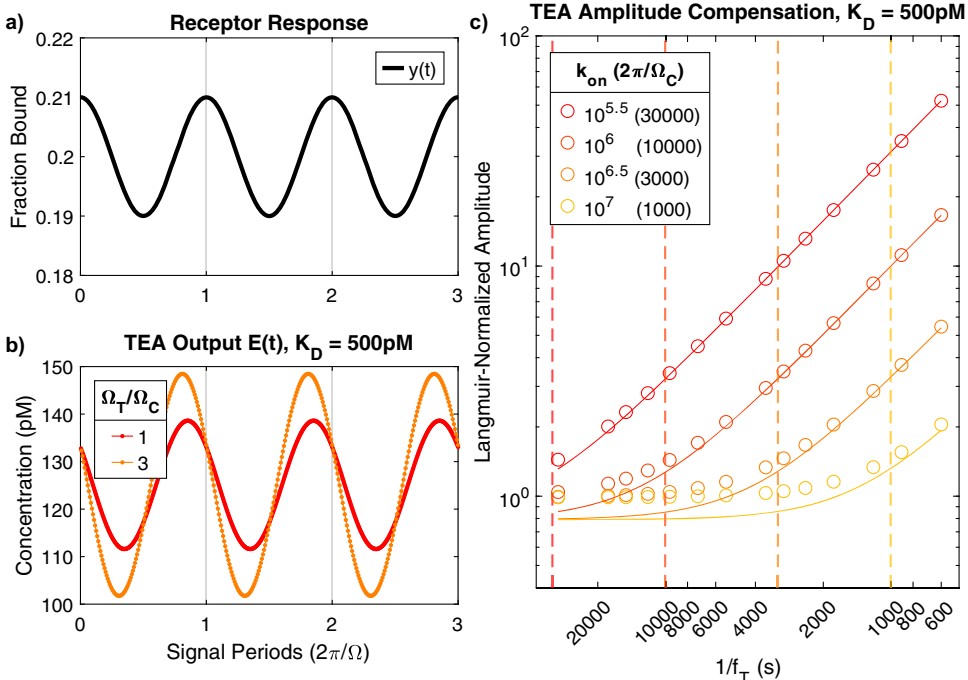

**Fig. 6 | Simulations of TEA. a** Receptor-binding signal $y(t)$ for a simulation with $C_0 = 0.2$ and $C_1 = 0.01$. **b** The signal was sampled and processed with the TEA (Eq. 7), where the receptor's kinetic parameters were $K_D = 500$ pM and $k_{on} = 10^6$ s$^{-1}$ M$^{-1}$. We performed this process with a fast-changing (high $\Omega_T/\Omega_C$; orange) and slow-changing (low $\Omega_T/\Omega_C$; red) signal to demonstrate that the TEA applies more compensation to faster-changing signals. **c** We repeated this simulation for a range of frequencies for four different receptors with equal $K_D$ but varying $k_{on}$ (and thus varying $\Omega_C$). $f_S = 1/30$ Hz. The solid lines depict Eq. 9, and circles show TEA-generated measurements of signal amplitude (Eq. 7) normalized to inverse-Langmuir predictions. Dashed vertical lines show $\Omega_C$ for each receptor.

fraction of bound receptor) to the samples. The resulting signal was then subjected to digital low-pass filtering. Because we aimed to resolve some of the sharper features of this signal—up to four times the fundamental frequency—we set the cutoff frequency of this filter at $f_T = 4/600$ Hz. This filter is effectively a weighted running average of the samples, which is designed to remove unwanted frequencies. The resulting samples $w[n]$ are shown in Fig. 7c. Finally, $w[n]$ was processed by the TEA to produce the target estimate $E[n]$ (Fig. 7d).

As the kinetics of the receptor became slower, the measurement was increasingly impaired by noise. The TEA applied stronger compensation to slower molecular systems, resulting in amplified noise at the output. We repeated this simulation for a wide range of receptor kinetics with a fixed $K_D$ of 500 pM and measured the output noise, defined as the error in the response of the system compared to a noiseless system (units of pM RMS), and the corresponding SNR. We compared these values with those computed through Eqs. 13 and 14 (Fig. 7e). The results indicate that $\Omega_T/\Omega_C < \sim 50$ (i.e., $k_{off} > \sim 10^{-3}$ s$^{-1}$) is required to distinguish the target signal from noise for the parameters simulated here (i.e., $T_0, T_1, K_D, f_T, \sigma_N$). Importantly, the results obtained through the noise scaling function (Eq. 13) are in good agreement with the measured values, indicating that our analytical equation for $S_N$ can be used to explore the design space and optimize the system as desired.

**Optimization in terms of the noise scaling function**

Considering Eq. 13 more closely, we observed that for a given $T_0$ and $k_{on}$ there is an optimal choice of $k_{off}$ (and consequently an optimal receptor $K_D$) that minimizes the amount of noise, and thus maximizes the SNR, of the estimated target concentration (Eq. 14). We explored this optimum for the insulin sensor case study in Fig. 8a, where we plot the noise scaling function, $S_N$, for two values of $k_{on}$, as well as the asymptotes of $S_N$ for large and small values of $k_{off}$. The location of the minimum value of $S_N$ is marked on the curves as well. As in the previous section, we used $f_T = 4/600$ Hz and $f_S = 10 f_T$. An insulin

sensor with $k_{on} = 10^{6.5}$ s$^{-1}$ M$^{-1}$ would achieve optimal noise performance with a $K_D \approx 1$ nM ($k_{off} \approx 3 \times 10^{-3}$ s$^{-1}$), while a sensor with $k_{on} = 10^{7.5}$ s$^{-1}$ M$^{-1}$ would be optimized with a $K_D \approx 300$ pM ($k_{off} \approx 10^{-2}$ s$^{-1}$ M$^{-1}$). Deviations from these optima tend to greatly increase noise. For example, for the case of $k_{on} = 10^{7.5}$ s$^{-1}$ M$^{-1}$, a receptor with a $K_D$ of 10 nM leads to 100-fold higher $S_N$ than using a $K_D$ of 1 nM. The same is true for values below the optimal $K_D$, showing that contrary to conventional thinking (and as demonstrated previously)[34], a lower $K_D$ is not always better. The optimal value of $K_D$ is plotted for a range of $k_{on}$ and $T_0$ in Fig. 8b. In our analysis of $S_N$ minima corresponding to optimal $K_D$, we noted a strong relationship between minimum $S_N$ and $k_{on}$ (Fig. 8c). For example, a 10-fold increase in $k_{on}$ reduced the minimum $S_N$ that could be achieved by 100-fold, indicating that the use of receptors with fast $k_{on}$ should offer an effective way of lessening the impact of noise on the sensor. We also note that for larger values of $T_0$, the minimum achievable $S_N$ does not scale as directly with $k_{on}$. This occurs in the regime where the optimal $K_D \leq T_0$. The implication is that measuring small concentration fluctuations ($T_1$) around a large mean concentration ($T_0$) is considerably more demanding on the noise requirement than measuring the same $T_1$ about a small $T_0$. We also found that increasing the number of signal harmonics resolved, in this case from 4 to 40 such that now $f_T = 40/600$ Hz ($f_S = 10 f_T$), is very costly in terms of noise performance, leading to a 1000-fold increase in the minimum achievable noise (Fig. 8b, c). This is primarily because increasing the bandwidth of interest results in less filtering and more noise in the output of the TEA. It is therefore critical to choose the minimum acceptable $f_T$ that will resolve the desired features of the signal. As previously mentioned, Lubken et al.[25] have provided an analysis of the frequency content of several important targets based on their endogenous concentrations—their method can be used to estimate an acceptable $f_T$.

As discussed above, it is critical in some cases to closely match the optimal $K_D$ in order to minimize noise. We propose a simple

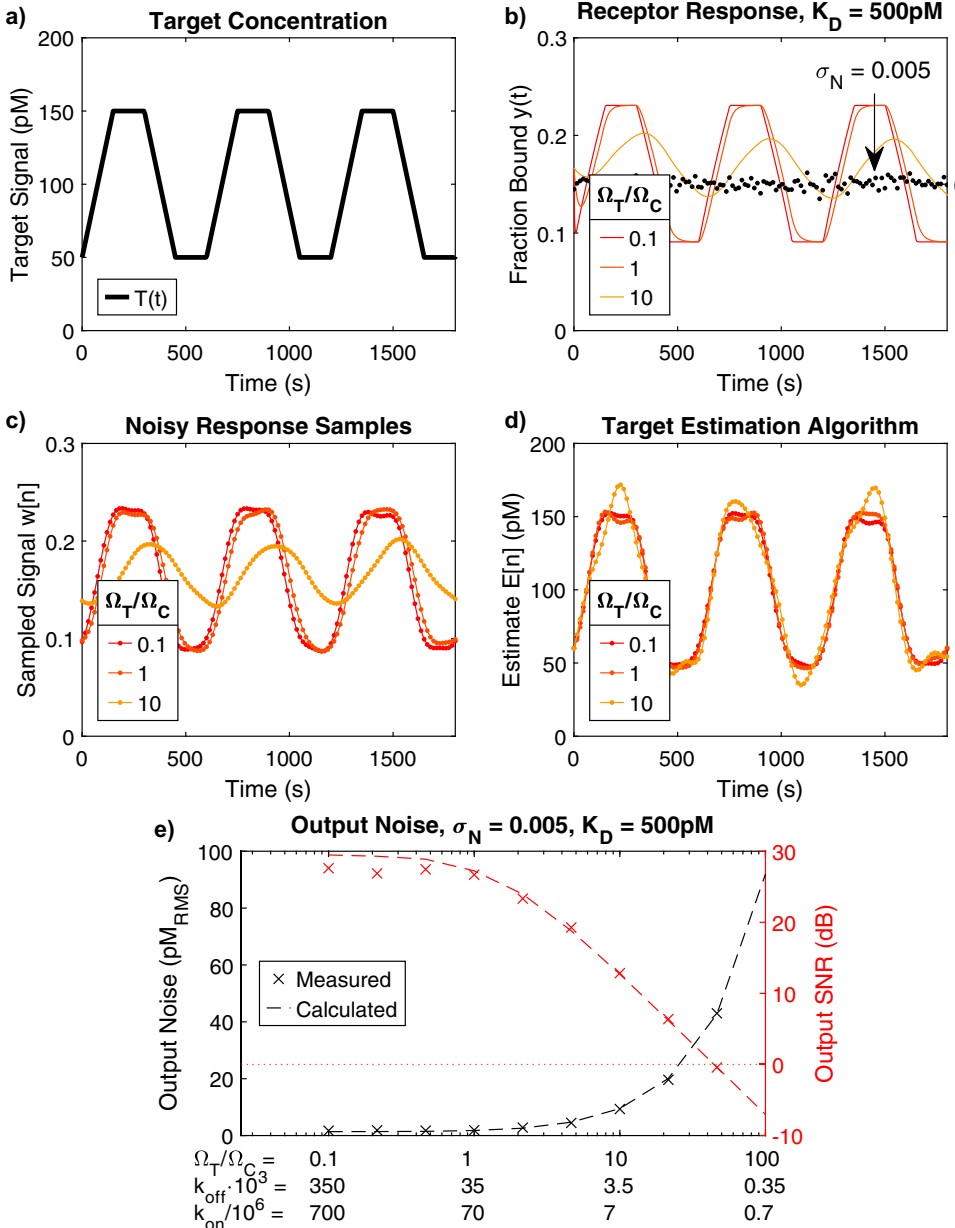

**Fig. 7 | End-to-end SNR simulation of insulin sensor. a** A trapezoidal test concentration signal with $T_0 = 100$ pM, $T_1 = 50$ pM, and a period of 10 min (1/600 Hz). **b** We simulated the response of three receptors with constant $K_D = 500$ pM but decreasing kinetics, where $\Omega_T/\Omega_C = 0.1$, 1, or 10. Also shown for comparison is the noise $N[n]$ introduced by the detector system (black dots). $N[n]$ is zero-mean white noise with a standard deviation of 0.005 fraction bound. **c** The receptor response was sampled with $f_S = 1/15$ Hz, combined with $N[n]$, and then filtered to remove all frequency content $>f_T$. Because we aimed to resolve four harmonics above the fundamental, we choose $f_T = 4/600$ Hz. **d** Estimated target concentration, $E[n]$, at the output of the TEA. **e** Simulations for systems with $K_D = 500$ pM over a range of $\Omega_T/\Omega_C$, where output noise and SNR were measured. Each simulation scenario is also reported in terms of the corresponding $k_{on}$ and $k_{off}$ values. Dotted lines depict Eq. 13 and Eq. 14.

approximation for the optimal $K_D$, valid when $f_S \gg f_T$:

$$K_D^\star \approx 1.90 \sqrt{\frac{f_T T_0}{k_{on}}}. \tag{15}$$

$K_D^\star$ approximates the receptor affinity at which the minimum noise point is achieved and can thus be used as a rule-of-thumb for identifying optimal operating conditions. This approximation is based on the location where the asymptotes shown in Fig. 8a intersect. The equations of the asymptotes are $S_N|_{k_{off} \to \infty} = \left(\frac{k_{off}}{k_{on}}\right)^2 f_T$ and $S_N|_{k_{off} \to 0} = \frac{T_0^2}{k_{off}^2}\left(2f_S^2 f_T - \frac{f_S^3}{\pi}\sin\left(\frac{2\pi f_T}{f_S}\right)\right)$. Their intersection point is $K_D^\star = \sqrt{T_0 \alpha/k_{on}}$, where the constant $\alpha^2 = 2f_S^2 - \frac{f_S^3}{\pi f_T}\sin\left(\frac{2\pi f_T}{f_S}\right)$. If $f_S \gg f_T$, $K_D^\star$ reduces to Eq. 15. We confirmed that the $S_N$ values corresponding to $K_D^\star$ are in close agreement with the actual minimum values of $S_N$ (Fig. 8c), and

even though $K_D^\star$ does not always accurately coincide with the optimal $K_D$, this disparity typically occurs when the minimum in $S_N$ is shallow.

Finally, we leveraged the cumulative insights from this work to describe the design parameters of an optimal pre-equilibrium real-time insulin sensor. We maintained the target concentration parameters cited above ($T_0 = 100$ pM, $T_1 = 50$ pM, $f_T = 4/600$ Hz, $f_S = 10f_T$). If we intend to design an insulin sensor with an output SNR of -14 dB, this would correspond to an average error of about 10 pM$_{RMS}$ for a 50 pM sinusoidal concentration signal. Based on Fig. 8, a receptor with $k_{on} = 10^6$ s$^{-1}$ M$^{-1}$ would perform optimally with $K_D \approx 3$ nM ($k_{off} \approx 3 \times 10^{-3}$ s$^{-1}$) and would achieve $S_N \approx 10^{-18}$ Hz$\left(\frac{moles}{fraction bound}\right)^2$. Using Eq. 12, we can calculate a noise specification for our detector:

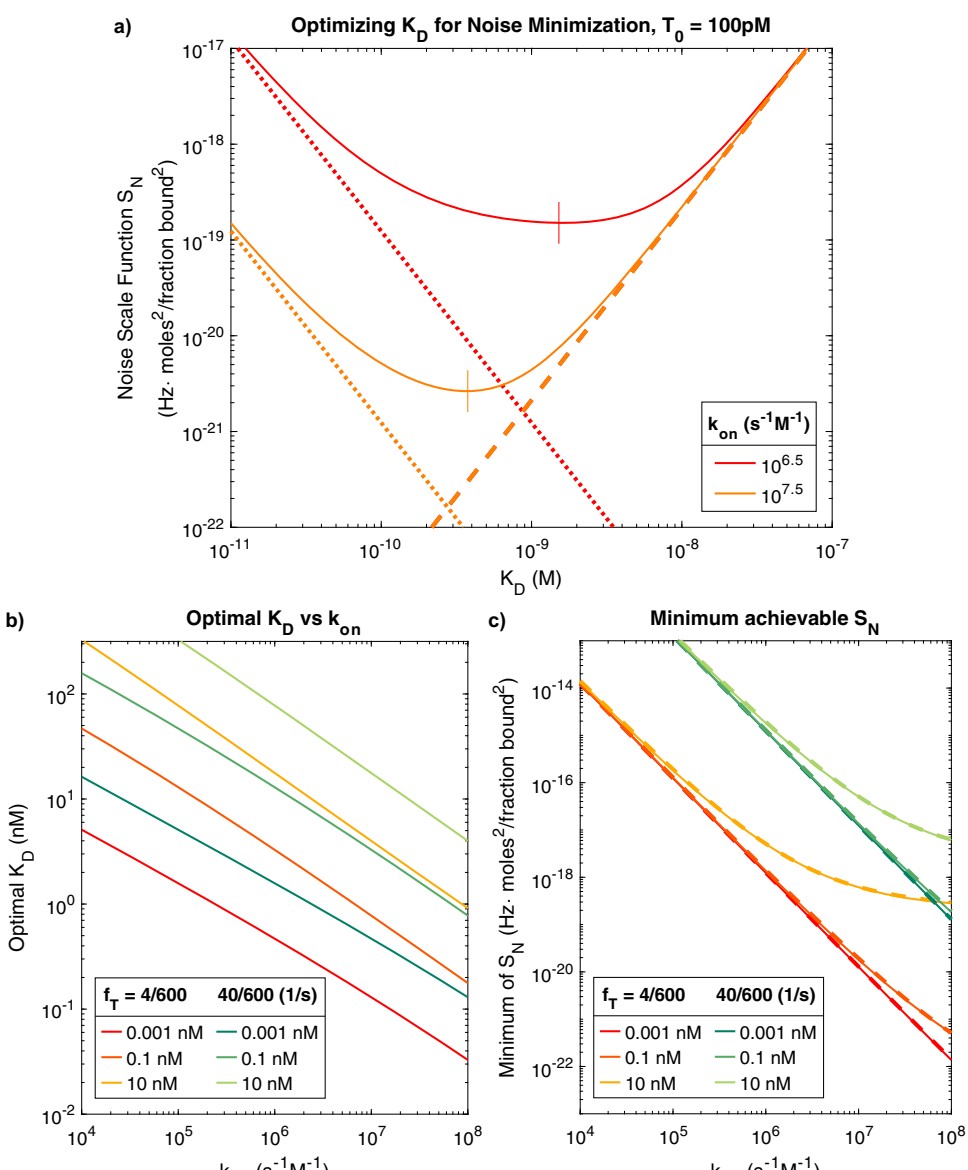

**Fig. 8 | Minimizing the noise scaling function ($S_N$). a** Investigating the optimal $K_D$ for two insulin receptors with $k_{on} = 10^{6.5}$ and $k_{on} = 10^{7.5}$, resolving an insulin concentration signal with $f_T = 4/600$ Hz, $f_S = 10f_T$, and $T_0 = 100$ pM. The noise scaling function (solid lines) achieves a minimum (vertical lines) at $K_D \approx 1$ nM for the slow receptor and $K_D \approx 300$ pM for the fast one. Also plotted are the asymptotes of $S_N$ for $k_{off} \to \infty$ (dashed lines) and for $k_{off} \to 0$ (dotted lines). **b** The optimal $K_D$ corresponding to the minimum $S_N$ was obtained for a range $k_{on}$ values for $T_0 = 0.001$, $0.1$, and $10$ nM for insulin sensors with $f_T = 4/600$ Hz (four harmonics, target period 600 s) and $f_T = 40/600$ Hz (40 harmonics, target period 600 s). **c** The minimum achievable $S_N$ (corresponding to optimal $K_D$ values) for the same conditions as b.

$N_0/2 \leq 10^{-4}$(fractionbound)$^2$/Hz. This corresponds to a noise standard deviation of $\sigma_N \leq \sqrt{f_S \cdot \frac{N_0}{2}} = 0.0026$ fraction bound, roughly half the one shown in Fig. 7b. Even for the demanding sensing requirements chosen for this example, we find that the kinetic parameters are within the expected range of typical antibodies[35,36] and that the noise requirement is tractable, based on the sensing performance of published biosensors[37-44]. This supports the idea that antibodies should be viable receptors for use in a real-time insulin sensor.

For purposes of comparison, we assessed the temporal resolution that could be achieved with a traditional real-time sensor that does not employ pre-equilibrium sensing and computes the target estimate by using the inverse-Langmuir alone. For the same insulin signal described above ($T_0 = 100$ pM, $T_1 = 50$ pM), an error equivalent to 10 pM$_{RMS}$ in the target estimate is introduced when the receptor response is attenuated by ~10% of its equilibrium value. Using Eq. 5, we find that this attenuation occurs at $\Omega_T = 1.5 \times 10^{-3}$ rad/s or $f_T = 2.39 \times 10^{-4}$ Hz

(i.e., a period of ~70 min). This frequency is the highest target frequency that a traditional equilibrium-based sensor could resolve with the same error as the pre-equilibrium sensor proposed above—note that this calculation is an upper-bound estimate as it neglects the impact of detector noise. Yet, the equilibrium sensor $f_T$ is ~28X slower than that of the pre-equilibrium sensor ($f_T = 4/600$ Hz, or 2.5 min) and demonstrates that our pre-equilibrium system offers the possibility of achieving greatly improved real-time monitoring of analytes with sensors based on existing bioreceptors.

To further demonstrate the real-world utility of the proposed method, we carried out a simulation that compares the performance of two receptors operated using either the pre-equilibrium TEA or the inverse-Langmuir alone. For this simulation, we used a smooth, aperiodic target concentration profile that more closely resembles a spike in insulin concentration (Fig. 9a). The simulated fraction-bound response of two receptors with fixed $k_{on} = 10^6$ s$^{-1}$M$^{-1}$ and $K_D = 1$ nM and 10 nM is shown in Fig. 9b. The receptor response was sampled at

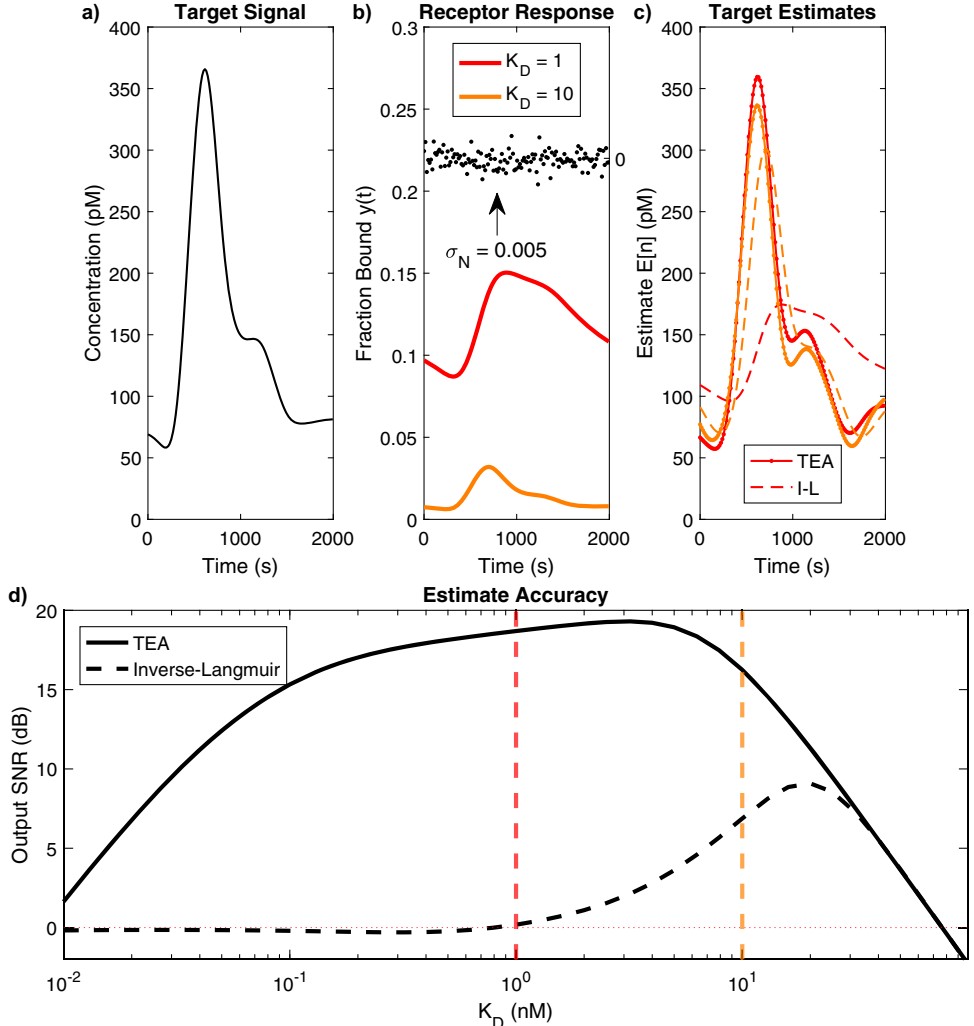

**Fig. 9 | A simulation of the impact of $K_D$ optimization using TEA versus inverse-Langmuir. a** Target concentration $T(t)$ used in this simulation. **b** Simulated response of two receptors with fixed $k_{on} = 10^6 \, s^{-1} \, M^{-1}$ but different $K_D$s to a spike in target concentration. Also shown for comparison is the noise $N[n]$ introduced by the detector system (black dots). In this simulation, $N[n]$ is zero-mean white noise with a standard deviation of 0.005 fraction bound. **c** The response of the receptors was sampled at $f_S = 1/15$ Hz and the noisy samples were digitally low-pass filtered with $f_T = 1/400$ Hz. The data was then used to reconstruct the original target concentration either using the pre-equilibrium TEA (solid lines) or conventional inverse-Langmuir (dashed lines) for the receptor with $K_D = 1$ nM (red) or 10 nM (orange). **d** We calculated the error in the target concentration after repeating the simulation with receptors with $k_{on} = 10^6 \, s^{-1} \, M^{-1}$ but a range of $K_D$s. This is shown in terms of SNR for both the pre-equilibrium TEA (solid line) and the simple inverse-Langmuir (dashed line). Vertical red and orange lines correspond to the simulations in (**b**) and (**c**).

$f_S = 1/15$ Hz, as in previous simulations. However, $f_T = 1/400$ Hz was sufficient to resolve this signal. Gaussian noise $N[n]$ with standard deviation $\sigma_N = 0.005$ fraction bound was added in the sampling process, and then a sharp digital filter with cutoff frequency $f_T$ was applied. The filtered data was then used to reconstruct the target concentration using both the pre-equilibrium TEA and the inverse-Langmuir (Fig. 9c). For both receptors, the TEA estimates were closer to the true target concentration (10.4 $pM_{RMS}$ and 13.8 $pM_{RMS}$ errors for the high and low-affinity receptors, respectively) than the inverse-Langmuir estimates (87.8 $pM_{RMS}$ and 40.6 $pM_{RMS}$). $K_D^\star$ (computed using $T_0 \approx 200$ pM) was ~1.3 nM; unsurprisingly, the TEA achieved greater accuracy with the $K_D = 1$ nM receptor than with the $K_D = 10$ nM receptor. Because the receptors never reached equilibrium, the inverse-Langmuir estimates had high levels of error and failed to capture the peak insulin concentration. Though the higher affinity receptor produced a larger fraction bound response, its inverse-Langmuir target estimate was less accurate than the one from the lower affinity receptor. This is due to the slower $k_{off}$ associated with higher affinity, which increases the error caused by the receptor's inability to reach equilibrium. We repeated

this simulation for a range of $K_D$s with fixed $k_{on} = 10^6 \, s^{-1} \, M^{-1}$, each time measuring the estimation error of the TEA and inverse-Langmuir. Figure 9c shows that the SNR of the reconstructed target concentration using the TEA was consistently higher than when using the inverse-Langmuir alone, with a peak SNR ~9.5 dB higher than the peak inverse-Langmuir SNR (a 10.5-fold increase in SNR), indicating that sensor designers should opt for the TEA algorithm over the inverse-Langmuir regardless of the choice of receptor. The TEA's peak SNR was well-aligned with the calculated value for $K_D^\star$, and the shape of the curve is in overall agreement with the optimization data shown in Fig. 8a.

In this work, we propose a method to measure real-time changes in target concentrations prior to target-receptor equilibration. While current sensing techniques discard valuable information embedded in the binding signal prior to receptor equilibration, pre-equilibrium sensing considers the dynamics of receptor equilibration to estimate the target concentration. Frequency-space analysis of pre-equilibrium sensing shows that bimolecular target-receptor interactions can be approximated as a first-order low-pass filter, i.e., the receptor

attenuates high-frequency content, and smooths out rapid changes in the concentration. To account for this, we propose a pre-equilibrium target estimation algorithm that reconstructs rapid changes in target concentration by compensating for the attenuations in high frequencies introduced by slow receptor kinetics. In an ideal noise-free system, this method exactly determines the target concentration at all points in time. Noise, however, makes it more challenging to measure concentration changes that are faster than the kinetics of the receptor because slower receptors require more compensation, which magnifies the noise in the target estimate. Thus, to enable informed design of pre-equilibrium systems, we derived closed-form expressions that relate parameters of the biosensor to the SNR of the estimated target concentration. We then created numerical simulations to study the performance of the biosensor system and found good agreement between simulation and our closed-form expressions. For the case of insulin sensing, we showed how these equations can be used to determine specifications for pre-equilibrium real-time sensors. Although the equilibration rate of antibodies is typically incompatible with the timescale of physiological insulin concentration changes, pre-equilibrium sensing can be leveraged to relax this requirement. We then verified this by simulating an insulin target concentrations and showed that the pre-equilibrium TEA vastly outperformed the inverse-Langmuir.

Interestingly, we found that the SNR of the target estimate in pre-equilibrium systems can be optimized in terms of the $K_D$ of the receptor—for a given receptor $k_{on}$, there is a specific $k_{off}$ that minimizes the error. Perhaps counterintuitively, in many cases the optimal $K_D$ falls at significantly larger concentrations than the average target concentration $T_0$. Conversely, a higher $k_{on}$ will often significantly reduce the error even when $k_{off}$ is left unchanged. Increasing $k_{on}$ is most impactful in the case of small average target concentrations, where the optimal $K_D$ falls above $T_0$. Also, $f_T$, i.e., the highest frequency in $T(t)$ that the designer is interested in measuring, is an impactful design parameter that can improve performance. For example, while a physiological signal might have (small) high-frequency fluctuations, these may not be of interest for the purposes of a sensing task. Choosing an $f_T$ that only encompasses features of the concentration signal that are of interest, perhaps eliminating small, high-frequency features, will dramatically reduce noise requirements of the sensing system. We believe that these design principles and analytical expressions can support the implementation of pre-equilibrium-based sensing strategies. While this work has primarily studied two-state bimolecular receptor-ligand interactions, future research could extend the theoretical insights to encompass more complex binding schemes, such as three-state receptors like aptamer switches[45]. Broadening the scope of pre-equilibrium techniques will improve biosensing performance in diverse applications by easing the requirement on receptor kinetics.

## Methods

### BLI experiment and characterization

For the characterization of antibody kinetics, a Sartorius Octet Red384 was used with anti-mouse Fc-capture biosensor tips (Sartorius). The tips were rehydrated in deionized (DI) water for 10 min. Anti-TNFα clones mAb1 and mAb11 were obtained from eBioscience (TNFa Mouse anti-Human Clone Name: mAb1, Supplier: Invitrogen, Catalog # 14734885; TNFa Mouse anti-Human Clone Name: mAb11, Supplier: Invitrogen, Catalog # 12734982). The antibodies were diluted to 90 nM in wash buffer (1X PBS, 0.1% Tween 20). The BLI sensor tips were dipped in antibody solution until the BLI shift was around 0.8 nm (about 60 s). The tips were then dipped in wash buffer for 180 s. Different sensor tips were then dipped in dilutions of 80 nM, 60 nM, 40 nM, 30 nM, 20 nM, and 0 nM recombinant TNFα (R&D Systems). The association curve was captured for 300 s, then the tips were moved to blank solution and the dissociation curve was observed for 900 s. ForteBio Octet DataAnalysis software was used to align and fit

the data (Supplementary Fig. 1a, b). The off-rate ($k_{off}$) was determined as the average of the $k_{off}$ values of each dissociation trace. The fitted association rates ($k_{obs}$) were then plotted vs target concentration (Supplementary Fig. 1c). The on-rate ($k_{on}$) was determined as the slope of the best fit line, $k_{obs} = k_{on}[T] + k_{off}$.

The data for the 20 nM experiment was truncated at 600 seconds and normalized by dividing all measurements by the RMax value provided by the analysis software (0.31654 nm for mAb1, 0.29594 nm for mAb11. This is the BLI shift value corresponding to a fraction bound equal to 1) after subtracting the baseline (0.004509 for mAb1, −0.01805 for mAb11). These normalized data are shown in Fig. 3a and were used to produce the results in Fig. 3b and c. The normalized data were downsampled to 1/15 Hz and digitally low pass filtered with a sharp cutoff at $f_T = 0.03$ Hz. The pre-equilibrium TEA was then applied, as described by Eq. 7.

The equilibrium estimates shown in Fig. 3b, c were calculated as follows. The equilibrium BLI shift values of the fitted association and dissociation curves were taken from the analysis software, normalized via RMax and baseline as described above to obtain equilibrium fraction bound values, and then the inverse-Langmuir was applied to compute the corresponding equilibrium concentration estimates. These values are shown in Supplementary Table 2.

A similar procedure was followed for the remaining data of Supplementary Fig. 1b and is included in Supplementary Fig. 2 as additional proof-of-principle data.

### Reporting summary

Further information on research design is available in the Nature Portfolio Reporting Summary linked to this article.

## Data availability

All data generated or analyzed during this study are included in this published article (and its supplementary information files), as well as at https://doi.org/10.5281/zenodo.7075990. Source data are provided with this paper.

## Code availability

All code is included in this published article's supplementary information files, as well as at https://doi.org/10.5281/zenodo.7075990.

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

## Acknowledgements

The authors would like to thank Mr. Michael Eisenstein and Vlad Kesler for their assistance editing drafts of the manuscript. N.M. and B.W. would like to acknowledge the support of the Chan-Zuckerberg Biohub, the Helmsley Trust, Bayer AG, and the National Institutes of Health (NIH, OT2OD025342). I.A.P.T. would like to acknowledge the support of the Medtronic Foundation Stanford Graduate Fellowship and the Natural Sciences and Engineering Research Council of Canada (NSERC) [funding reference number 416353855].

## Author contributions

N.M., I.A.P.T., B.D.W., and H.T.S. conceived the initial concepts. N.M. performed the mathematical analysis, developed the model, performed the experiments, and analyzed the data. I.A.P.T. performed the mass transport and Poisson noise analysis. N.M. and H.T.S. wrote the manuscript. All authors edited, discussed, and approved the whole paper.

## Competing interests

The authors declare no competing interests.
