## [Peer Review File · Nature Communications]

REVIEWER COMMENTS

Reviewer #1 (Remarks to the Author):

The manuscript by Maganzini et al. describes a method to process signals of a continuous biosensor based on reversible receptor-ligand interactions. Such biosensors behave as low pass filters, attenuating signals at high frequencies. The manuscript proposes to amplify high frequency signals using a target estimation algorithm (TEA), in order to compensate for signal losses related to the limited speed of receptor-ligand interactions (see figure 3b). However, a signal compensation approach is not trivial as it also introduces errors and increases errors. Therefore detailed understanding is required of random and systematic errors in the biosensing approach.

Some observations and remarks:

1. The manuscript is only theoretical. To convince readers about impact, it is essential to also obtain experimental support. Experiments can easily be done, for example using a label free sensing method such as SPR, QCM, or BLI.
2. The model neglects transport effects, such as diffusion and advection. Transport effects are always present in biosensors. Furthermore, transport effects cause time- and frequency-dependent behavior. Therefore, transport effects are relevant for the central question of this manuscript. The role of transport effects should be quantitatively discussed in the manuscript.
3. The manuscript assumes white noise for the sensor. It neglects Poisson noise without giving any explanation. Poisson noise is always present in molecular sensors, so this topic should be treated quantitatively.
4. The manuscript says "... we assume that solution-phase target molecules are in great excess relative to the sensor's receptors". This is not a trivial assumption, because the number of target molecules that can interact with the sensor relates to concentrations, the timescale of the biosensing assay, and the transport effects (diffusion). The manuscript should include a discussion on when the assumption holds and when not.
5. Random errors lead to concentration imprecision; systematic errors lead to concentration inaccuracy. The manuscript focuses on noise, i.e., on imprecision. However, signal amplification will also increase systematic errors and cause inaccuracies, which should be discussed in the manuscript. The manuscript should make clear distinctions between concentration measurement imprecision and concentration measurement inaccuracy.
6. The manuscript focuses on "a sensor for insulin tracking". However, the simulations use periodic model profiles (sinusoidal). The manuscript should also include realistic concentration-time profiles for insulin, which are not periodic.

7. The manuscript compares sensor performance for sensors with different affinity parameters. I think a figure should be added that plots the concentration measurement imprecision and inaccuracy of the biosensor as a function of K_d , showing curves with as well as without TEA applied. This will help readers to understand the advantages of different approaches (e.g. change K_d versus apply signal processing strategy).

8. The impact of applying the TEA is insufficiently clear to me. Will the method lead to fundamentally new levels of sensing functionality that are widely applicable, or are the improvements incremental and suited for a limited biosensor scope? This requires quantitative extrapolations based on theoretical as well as experimental results.

Given the above, I think the present manuscript cannot be accepted for publication.

Other points:

. The section between “Principles and limitations of equilibrium sensing” until “Frequency response of the receptor” seems superfluous. I think this section can at least partially be moved to supplementary information.

. The conclusion states that the optimal K_d is larger than T_0 . This seems according to expectation, because a sensor with $K_d < T_0$ approaches saturation and becomes insensitive to concentration fluctuations. Is this viewpoint correct?

Reviewer #2 (Remarks to the Author):

This manuscript presents an analysis of the temporal (non-equilibrium) response of sensors based on receptor-ligand binding, motivated by the kinetic limitations of biosensors for sensing time-variant concentrations in cases such as insulin monitoring. The analysis is performed with the key assumptions that the sensor does not perturb the target concentration (equivalently, that mass transport is not limiting), and that the sensor has a set gaussian noise in its readout that is independent of the receptor used. The results are intriguing and outline how to design a sensor and select receptors for measuring time-variant target concentrations. Although the manuscript does not provide experimental validation or involve new kinds of analyses, it proposes a new approach to biosensing of time-varying target concentrations and provides a comprehensive basis for the design of such biosensors. One example of an important but counterintuitive result is that there is an optimal affinity for measuring time-variant target concentrations, and receptors with higher affinity (normally considered better for sensing) can actually perform much worse than lower-affinity receptors. . It is likely that this work will inspire a new class of biosensors for real-time in vivo measurement, and possibly sensing in other applications as well. The manuscript is well-organized and the reviewer recommends publication after addressing the minor comments below.

Page 3, line 38 – It may be helpful to add a sentence on how the timescale for equilibration depends on k_{on} and k_{off} to close the argument being made (it is stated on line 114 for the condition where the number of receptors is small compared to number of targets). A brief discussion of mass transport effects that play a role when the receptors bind and deplete the targets would also be helpful, either on page 3 or on page 5.

Page 5, line 94, and page 7, lines 124-126 – is the assumption that the number of target molecules bound is sufficiently high that the noise in y is negligible typically a valid one? Can briefly state the assumption and under what conditions it is valid. It will be helpful to provide an expression for the validity of the assumption that the target concentration is unaffected by the receptors. The assumption should be OK when the timescale for equilibration ($1/k_{eq}$) is larger than the timescale for diffusion across a volume given by the number of targets bound to the receptors at equilibrium divided by the target concentration.

Page 11, line 222 – it would be helpful for the sake of clarity to add an equation stating that the measured signal is given by $w[n] + \text{noise}$, and that $w[n]$ depends on y by some constant linear scaling factor.

Page 12, line 230 – need to briefly state the key assumptions in the analysis here.

It will be helpful to clarify somewhere that the same sensor can be functionalized with different receptors. This will justify the assumption that the sensor noise is independent of the receptor properties. A sketch in figure 1 or a figure before figure 1 may be helpful.

Adding a glossary of symbols will be helpful.

Reviewer #4 (Remarks to the Author):

This work on biosensor is interesting and seems to be very effective tool for detection. This work provides a theoretical foundation for a novel method to achieve real-time analyte quantification with biomolecular receptors at pre-equilibrium seems very robust technique of biomolecules detection at low concentration. I would like to make some comments as follows:-

"Since authors describe this work as to achieve sensing ability of the device before the equilibrium is reached, that is, in pre-equilibrium state (non-equilibrium process), therefore, I would like to know is this process thermodynamically favorable? What about the Gibbs free energy change for this pre-equilibrium process?"

A receptor with fast kinetics will respond equally well to low- and high-frequency content. How can we enhance the kinetic of the receptor for real application following of this strategy?

Conclusions should be written concisely, exploring key findings of the study

Response Letter

Reviewer 1:

We thank the reviewer for the careful examination of our work and the thoughtful and insightful comments and suggestions. We address these below.

1. The reviewer requested experimental support for our theoretical work: “The manuscript is only theoretical. To convince readers about impact, it is essential to also obtain experimental support. Experiments can easily be done, for example using a label free sensing method such as SPR, QCM, or BLI.”

As suggested by the reviewer, we have conducted BLI experiments and applied our pre-equilibrium analysis technique to demonstrate its usefulness. We show real-time binding data for two antibodies with different kinetics and thermodynamics and show how the TEA algorithm reconstructs the target concentration instantaneously. We have also compared this to the standard inverse Langmuir to show the performance improvement obtained with the pre-equilibrium method. The following text appears in the revised manuscript:

Validation of pre-equilibrium sensing through a biolayer interferometry experiment

To validate the working principle of pre-equilibrium measurement, we carried out a biolayer interferometry (BLI) experiment (methods in Appendix 2). We measured the binding signals of two monoclonal antibodies – mAb1 ($k_{on} = 3.8 \times 10^5 \text{ s}^{-1}\text{M}^{-1}$, $k_{off} = 2.48 \times 10^{-3} \text{ s}^{-1}$) and mAb11 ($k_{on} = 1.4 \times 10^5 \text{ s}^{-1}\text{M}^{-1}$, $k_{off} = 2.10 \times 10^{-3} \text{ s}^{-1}$) – that were exposed to a step in target concentration: 20 nM followed by 0 nM TNF α (**Fig. 3a**). We then applied pre-equilibrium estimation as well as the inverse-Langmuir to estimate the target concentration at every instant in time (**Fig. 3b, c**). We found that, although the BLI sensor did not fully reset when exposed to the blank solution (possibly due to non-specific binding), the proposed pre-equilibrium method achieved much more rapid target quantification compared to the inverse-Langmuir.

During this experiment, the sensor did not reach equilibrium due to insufficiently rapid binding kinetics. However, for purposes of comparison, the raw sensor data was fitted to association and dissociation exponential curves using the BLI analysis software and extrapolated to determine the fraction bound at equilibrium (see Appendix 2). We used these values to determine the hypothetical inverse-Langmuir target estimate, had the receptor kinetics been fast enough to reach equilibrium. These estimates are shown in **Figure 3b** and **c** as solid black lines.

We note that there is some discrepancy between the true target concentrations used and the equilibrium estimates, particularly in the second step, where the sensor does not equilibrate back to 0 fraction bound.

This data demonstrates how the pre-equilibrium analysis can reach the same estimate as the inverse-Langmuir without the need of equilibration. As would be done in a real biosensor, the noisy sensor data was digitally low-pass filtered to remove as much noise as possible while preserving the signal shape (**Fig. 3a**, black lines). First, the inverse-Langmuir was applied to this filtered data to show that, because the receptor's kinetics are slower than the target concentration changes, the instantaneous concentration estimated by the inverse-Langmuir (**Fig. 3b, c**, dotted lines) is highly inaccurate. Conversely, using the same sensor data, the proposed pre-equilibrium method, which leverages Eq. **Error! Reference source not found.**, instantaneously estimated a target concentration that was much closer to the equilibrium estimate, for both the faster and slower receptors (**Fig. 3b, c**, solid lines). However, it should be noted that the pre-equilibrium estimates display noticeable variance about their mean, which reduces the precision of the measurement. As introduced in the previous section, the precision of pre-equilibrium measurements depends on the amount of noise in the detector relative to the rapidity of the receptor's kinetics. Conversely, low-frequency offsets such as the one shown in this experiment, impact pre-equilibrium and inverse-Langmuir estimates equally. In the analysis that follows we explore and characterize the frequency dependence of pre-equilibrium estimation in the presence of noise to inform the design of sensor systems that leverage this technique.

Figure 3: Experimental proof of principle. **a)** Normalized biolayer interferometry (BLI) binding data for two monoclonal antibodies, mAb1 (red) and mAb11 (orange), exposed to the following TNF α concentrations: from 0–300s, 20 nM, and from 300–600s, 0 nM. Black lines represent digitally low-pass filtered data. The target concentration was estimated using the pre-equilibrium TEA (solid lines) as well as the conventional inverse-Langmuir (dashed lines) for **b)** mAb1 and **c)** mAb11 data. Solid black lines represent the inverse-Langmuir target estimates based on the fraction of bound receptors at equilibrium.

Below, we report Appendix 2, which is also relevant to the reviewer's comment.

Appendix 2: BLI experiment and characterization

For the characterization of antibody kinetics, a Sartorius Octet Red384 was used with anti-mouse Fc-capture biosensor tips (Sartorius). The tips were rehydrated in deionized (DI) water for 10 minutes. Anti-TNF α clones mAb1 and mAb11 were obtained from eBioscience. The antibodies were diluted to 90 nM in wash buffer (1X PBS, 0.1% Tween 20). The BLI sensor tips were dipped in antibody solution until the BLI shift was around 0.8 nm (about 60 s). The tips were then dipped

in wash buffer for 180 s. Different sensor tips were then dipped in dilutions of 80 nM, 60 nM, 40 nM, 30 nM, 20 nM, and 0 nM recombinant TNF α (R&D Systems). The association curve was captured for 300 s, then the tips were moved to blank solution and the dissociation curve was observed for 900 s. ForteBio Data Analysis software was used to align and fit the data (**Fig. S1a,b**). The off-rate (k_{off}) was determined as the average of the k_{off} values of each dissociation trace. The fitted association rates (k_{obs}) were then plotted vs target concentration (**Fig. S1c**). The on-rate (k_{on}) was determined as the slope of the best fit line, $k_{obs} = k_{on}[T] + k_{off}$.

Figure S1: BLI Characterization. a) mAb1 and b) mAb11 association and dissociation data, fitted with the ForteBio Data Analysis software. c) Fit of k_{obs} vs target concentration used to determine k_{on} .

The data for the 20 nM experiment was truncated at 600 seconds and normalized by dividing all measurements by the RMax value provided by the analysis software (0.31654 nm for mAb1, 0.29594 nm for mAb11). This is the BLI shift value corresponding to a fraction bound equal to 1) after subtracting the baseline (0.004509 for mAb1, -0.01805 for mAb11). These normalized data are shown in **Figure 3a** and were used to produce the results in **Figure 3b** and **c**. The normalized data were downsampled to 1/15 Hz and digitally low pass filtered with a sharp cutoff

at $f_T = 0.03$ Hz. The pre-equilibrium TEA was then applied, as described by Eq. **Error!**
Reference source not found..

The equilibrium estimates shown in **Figure 3b** and **c** were calculated as follows. The equilibrium BLI shift values of the fitted association and dissociation curves were taken from the analysis software, normalized via RMax and baseline as described above to obtain equilibrium fraction bound values, and then the inverse-Langmuir was applied to compute the corresponding equilibrium concentration estimates. These values are shown in **Table S2**.

Table S2: Calculation of target estimates based on BLI equilibrium fraction bound.

	mAb1	mAb11
BLI Shift at equilibrium 20 nM - association	0.2613 nm	0.1508 nm
Fraction bound at equilibrium 20 nM - association	0.8112	0.5706
Inverse-Langmuir at equilibrium 20 nM - association	28 nM	20 nM
BLI Shift at equilibrium 0 nM - dissociation	0.2185	0.07245
Fraction bound at equilibrium 0 nM - dissociation	0.6760	0.3058
Inverse-Langmuir at equilibrium 0 nM - dissociation	13.5 nM	7 nM

2. The reviewer requested a quantitative discussion of transport effects such as diffusion and advection: “The model neglects transport effects, such as diffusion and advection. Transport effects are always present in biosensors. Furthermore, transport effects cause time- and frequency-dependent behavior. Therefore, transport effects are relevant for the central question of this manuscript. The role of transport effects should be quantitatively discussed in the manuscript.”

We thank the reviewer for raising this point. Mass transport effects were excluded from the model primarily because a detailed frequency-space analysis (analogous to the one presented in our work) of biosensors affected by mass transport effects has already been proposed in publications by Lubken *et al.* Their work treats mass transport delays and reaction kinetics as two separate systems and their analysis looks at the frequency response of each system individually and combined. The focus of our work is not so much on extremely low concentration analytes, where diffusion effects are critical. Rather, our work focuses on cases where the kinetic properties of the receptors are significantly mismatched to the concentration change rates of the analytes of interest.

To better convey these concepts to the reader, we have added a new Appendix 1, in which we clarify the assumptions regarding mass transport delays, and quantitatively describe the regimes where these assumptions are valid. We also include additional analysis to address comments 3 and 4 as well. The relevant section of Appendix 1 here reads as follows:

Beyond receptor properties, analyte transport to the sensor surface is an important consideration in examining the kinetics of biosensors. For our proposed pre-equilibrium model, we assume that the underlying sensor operates in a reaction-limited regime – that is, the overall sensor response is limited by the kinetics of the receptor rather than by diffusive and advective analyte transport to the sensor surface. In this reaction-limited regime, analyte transport has a negligible effect on the dynamics of the sensor response, which will then be solely governed by the law of mass-action described in Eq. **Error! Reference source not found.** and **Error! Reference source not found.** Fortunately, there are established methods for designing biosensors to ensure they operate in this regime; thus, our approach can be applied in a broad range of applications as long as these methods are considered.

Squires *et al.*⁴⁵ have provided a comprehensive framework for analyzing the interplay between analyte transport and reaction in surface-based biosensors, along with design principles based on this analysis. They describe the Damkohler number, Da , that determines if a sensor with given properties (*e.g.* geometry, sample flow-rates, surface and receptor preparation) acts within a reaction-limited or transport-limited regime. If $Da \gg 1$, the sensor acts in the transport-limited regime, whereas if $Da \ll 1$ the sensor is reaction-limited and thus in the regime where our analysis holds. Da is computed as

$$Da = \frac{k_{on} b_m L}{D \mathcal{F}} \quad (1)$$

where k_{on} is the receptor on-rate, b_m is the density of receptors on the sensor surface, and D is the diffusivity of the analyte. This model assumes the biosensor takes the form of a planar surface exposed to continuous flow of sample inside a measurement chamber with a rectangular cross-section, where L is the length of the sensor and \mathcal{F} is the dimensionless flux (which is computed based on the sensor length and width, sensing channel height and width, and sample flow-rate).

Using this framework, it is straightforward to design sensors that operate in a reaction-limited regime, such that our pre-equilibrium estimation approach can be applied.

As an example of this design method, we consider a microsensor of dimensions $50 \times 50 \mu\text{m}$ that is supplied with sample at a flow rate of $100 \mu\text{L}/\text{min}$ – a reasonable flow-rate for human biofluids such as blood. For a representative sensor based on our proposed optimized insulin-sensing system (**Table S1**), we find that $\text{Da} = 0.129$, and thus the sensor operates very close to the purely reaction-limited regime ($\text{Da} \ll 1$). The ratio between the overall time constant of sensor equilibration (τ_{CRD}), which incorporates effects of convection, reaction, and diffusion, and time constant of the reaction alone (τ_R) will be bounded by $\frac{\tau_{CRD}}{\tau_R} \leq 1 + \text{Da} = 1.129$. In other words, we can expect no more than a 12.9% slowdown in sensor response. Importantly, these assumptions are conservative with respect to mass transport, as the model assumes a purely laminar flow. In many cases, transport to the surface could be further enhanced – for example, using chaotic mixing by herringbone microfluidic structures^{27–30} – to further drive the sensor into a reaction-limited regime where analyte transport to the sensor poses no limitation to the kinetics of sensing.

Table S1: Representative values of sensor properties used in calculation of Poisson noise and Damkohler number calculations

Sensor Parameter	Representative Value	Notes
Receptor on-rate k_{on}	$10^6 \text{ s}^{-1}\text{M}^{-1}$	Value used in the insulin sensing model in Figure Error! Reference source not found.. Note this value is relatively high for an Ab and is thus conservative with respect to the reaction-limited assumption.
Receptor surface density b_m	$1 \cdot 10^{12} \text{ cm}^{-2}$	Approximate upper limit of surface density for antibody immobilization ⁴⁵ , as well as conventional surface density for aptamer immobilization ⁴⁶ .
Sensor length L	$50 \mu\text{m}$	Typical dimensions for a microsensor.
Sensor width W_s	$50 \mu\text{m}$	

Measurement channel width W_C	50 μm	
Measurement channel height H	100 μm	
Diffusivity D	100 $\mu\text{m}^2 \text{ s}^{-1}$	Diffusivity for insulin in aqueous solution ⁴⁷ .
Target concentration T_0	150 pM	The maximum insulin concentration from our insulin sensing model in Figure Error! Reference source not found. . The highest concentration will give the most conservative estimate for shot noise (i.e. , the largest shot noise amplitude).
Receptor affinity K_D	3 nM	Value used in the insulin sensing model in Figure Error! Reference source not found. .

3. The reviewer requested a quantitative treatment of the Poisson noise that is always present in molecular sensors: “The manuscript assumes white noise for the sensor. It neglects Poisson noise without giving any explanation. Poisson noise is always present in molecular sensors, so this topic should be treated quantitatively.”

The reviewer is correct that Poisson noise is always present in biosensors. However, the magnitude of this noise source scales inversely to the square root of the number of sampled receptors and is thus very small for many types of biosensors. For example, for micron or millimeter-scale sensor surfaces performing ensemble measurements, the shot noise is typically negligible. As stated in the manuscript, our work assumes that the detector is sampling a large number of receptors such that white noise dominates. To better convey this idea to the reader, we have quantitatively compared shot noise levels and gaussian noise levels for the sensor we describe in Appendix 1, and find that the former is over two orders of magnitude smaller than the latter. The relevant section of Appendix 1 reads as follows:

Another fundamental limitation on sensor design is imposed by the presence of Poisson noise (also termed shot noise) due to statistical fluctuations in the number of bound receptors on the sensor surface. At a given target concentration, the number of bound receptors on a sensor

surface follows a Poisson distribution, where the mean number of bound receptors, $\overline{N_{bound}}$, is given by the Langmuir isotherm as

$$\overline{N_{bound}} = \frac{T_0}{T_0 + K_D} \cdot N_{total} = \frac{T_0}{T_0 + K_D} \cdot b_m L W_s \quad (2)$$

where the total number of receptors (N_{total}) can be inferred from the length (L), width (W_s) and receptor surface density (b_m) of the sensor. The standard deviation of a Poisson distribution is the square root of its mean, and thus the shot noise in terms of bound receptors is $N_{shot} = \sqrt{\overline{N_{bound}}}$.

To determine if this additional Poisson noise will act as a significant factor in our system, we need to compare its amplitude to the amplitude of the zero-mean white noise that we assume to be the dominant noise in our model. This white noise has a standard deviation of $\sigma_N = 5 \cdot 10^{-3}$ fraction bound, and thus to make a comparison, we calculate the standard deviation of our Poisson distribution in terms of fraction bound as:

$$\sigma_{shot} = \frac{N_{shot}}{N_{total}} = \frac{\sqrt{\frac{T_0}{T_0 + K_D} \cdot b_m L W_s}}{b_m L W_s} = \sqrt{\frac{T_0}{(T_0 + K_D) b_m L W_s}} \quad (3)$$

This equation indicates that larger or denser sensors with more receptors will experience less fraction bound shot noise, as statistical fluctuations in bound receptors become less significant. Applying this calculation to the representative microsensor parametrized in our analysis of analyte transport to the sensor surface, we find the shot noise to be $\sigma_{shot} = 4.4 \cdot 10^{-5}$. This is much smaller than our assumed white noise ($\sigma_N = 5 \cdot 10^{-3}$), and thus can be assumed to be negligible in our analysis.

4. The reviewer Asked us to clarify when our assumption holds true and when it does not: “ ‘... we assume that solution-phase target molecules are in great excess relative to the sensor’s receptors’. This is not a trivial assumption, because the number of target molecules that can interact with the sensor relates to concentrations, the timescale of the biosensing assay, and the transport effects (diffusion). The manuscript should include a discussion on when the assumption holds and when not.”

We have elaborated on this assumption in the following passage of Appendix 1:

By ensuring that our sensor acts in the reaction-limited regime, we also enforce our assumption that solution-phase analyte is in great excess relative to receptors on the sensor surface. When $Da \ll 1$, the flux of analyte to the sensor surface through advection and diffusion is much faster than the rate of binding, offsetting any local depletion of target caused by sensor binding. As long as there is sufficient sample volume to maintain continuous flow across the sample surface, receptors on the surface will never deplete the solution, and analytes will always remain in vast excess. For example, in the scenario described above, the number of targets found within just ~ 13 nL of solution containing 150 pM of analyte will be sufficient to occupy the bound fraction of receptors present on the sensor surface. Thus, equilibration without depletion can be readily achieved with the 100 $\mu\text{L}/\text{min}$ flow-rate utilized in the model. For continuous sensing in human biofluids such as blood, where sampling milliliter volumes is possible, these continuous flow-rates are readily attainable.

5. The reviewer noted that “Random errors lead to concentration imprecision; systematic errors lead to concentration inaccuracy. The manuscript focuses on noise, i.e., on imprecision. However, signal amplification will also increase systematic errors and cause inaccuracies, which should be discussed in the manuscript. The manuscript should make clear distinctions between concentration measurement imprecision and concentration measurement inaccuracy.”

We respectfully disagree with the reviewer’s comment. Although systematic error is present in biosensors, such errors affect equilibrium and pre-equilibrium biosensors equally. This is because, by definition, systematic errors are zero-frequency (such as instrumentation offsets) or very low frequency (such as sensor drift). Because the TEA algorithm only amplifies high frequency content in the sensor readout, such low frequency noise components would only affect the inverse-Langmuir (see Eq. 9 and 10). The same would be the case for the sensor operating at equilibrium. We have demonstrated this in the new experimental data included in the manuscript, where the effect of non-specific binding is the same for both the TEA and the inverse-Langmuir. We have added the following section to the text, discussing how fraction-bound signal is obtained from raw sensor readout:

Importantly, throughout this work, we assume that the sensor readout $w[n]$ is in units of fraction bound, with values in the range of 0 to 1. To achieve this, a real-world sensor system requires calibration of the raw detector readout (in units of current, voltage, fluorescence intensity, etc.) to determine its correlation with fraction bound. This often involves normalization and background subtraction, and inaccuracies in this process may introduce systematic errors in the readout, which will increase the error in the estimation of target concentrations. Systematic errors may also be introduced by phenomena such as sensor drift and offsets due to non-specific binding. As will be shown later, these sources of inaccuracy affect pre-equilibrium and equilibrium sensors equally due to their low-frequency nature. Therefore, this work will instead focus on analyzing the impact of random noise introduced during measurement, which impacts the proposed pre-equilibrium estimation method more significantly than it does conventional inverse-Langmuir estimation.

6. The reviewer requested more realistic (non-periodic, smooth) concentration-time profiles for insulin: “The manuscript focuses on ‘a sensor for insulin tracking’. However, the simulations use periodic model profiles (sinusoidal). The manuscript should also include realistic concentration-time profiles for insulin, which are not periodic.”

We agree that a more realistic insulin profile would help the reader to better appreciate the contributions of this work. We have therefore included an additional simulation that implements the optimized insulin receptor we describe in tracking an insulin concentration signal. The simulation also compares an optimized receptor to one with a non-optimal K_D , where both are operating in pre-equilibrium. We evaluate the performance of these receptors by comparing them to equivalent equilibrium sensors that estimate the target concentration using the inverse-Langmuir alone. We then plot the estimation error of both the TEA and the inverse-Langmuir over a range of receptor K_D values to show how selecting the optimal reduces the estimation error. These results are described as follows:

To further demonstrate the real-world utility of the proposed method, we carried out a simulation that compares the performance of two receptors operated using either the pre-equilibrium TEA or the inverse-Langmuir alone. For this simulation, we used a smooth, aperiodic target concentration profile that more closely resembles a spike in insulin concentration (**Fig. 9a**). The simulated fraction bound response of two receptors with fixed $k_{on} = 10^6 \text{ s}^{-1}\text{M}^{-1}$ and $K_D = 1 \text{ nM}$ and 10 nM is shown in **Figure 9b**. The receptor response was sampled at $f_S = 1/15 \text{ Hz}$, as

in previous simulations. However, $f_T = 1/400$ Hz was sufficient to resolve this signal. Gaussian noise $N[n]$ with standard deviation $\sigma_N = 0.005$ fraction bound was added in the sampling process, and then a sharp digital filter with cutoff frequency f_T was applied. The filtered data was then used to reconstruct the target concentration using both the pre-equilibrium TEA and the inverse-Langmuir (**Fig. 9c**). For both receptors, the TEA estimates were closer to the true target concentration (10.4 pM_{RMS} and 13.8 pM_{RMS} errors for the high and low affinity receptors, respectively) than the inverse-Langmuir estimates (87.8 pM_{RMS} and 40.6 pM_{RMS}). K_D^* (computed using $T_0 \approx 200$ pM) was approximately 1.3 nM; unsurprisingly, the TEA achieved greater accuracy with the $K_D = 1$ nM receptor than with the $K_D = 10$ nM receptor. Because the receptors never reached equilibrium, the inverse-Langmuir estimates had high levels of error and failed to capture the peak insulin concentration. Though the higher affinity receptor produced a larger fraction bound response, its inverse-Langmuir target estimate was less accurate than the one from the lower affinity receptor. This is due to the slower k_{off} associated with higher affinity, which increases the error caused by the receptor's inability to reach equilibrium. We repeated this simulation for a range of K_D s with fixed $k_{on} = 10^6$ s⁻¹M⁻¹, each time measuring the estimation error of the TEA and inverse-Langmuir. **Figure 9c** shows that the SNR of the reconstructed target concentration using the TEA was consistently higher than when using the inverse-Langmuir alone, with a peak SNR ~ 9.5 dB higher than the peak inverse-Langmuir SNR (a 10.5-fold increase in SNR), indicating that sensor designers should opt for the TEA algorithm over the inverse-Langmuir regardless of the choice of receptor. The TEA's peak SNR was well-aligned with the calculated value for K_D^* , and the shape of the curve is in overall agreement with the optimization data shown in **Figure Error! Reference source not found.a**.

Figure 9: A simulation of the impact of K_D optimization using TEA versus inverse-Langmuir. a) Target concentration $T(t)$ used in this simulation. **b)** Simulated response of two receptors with fixed $k_{on} = 10^6 \text{ s}^{-1}\text{M}^{-1}$ but different K_D s to a spike in target concentration. Also shown for comparison is the noise $N[n]$ introduced by the detector system (black dots). In this simulation, $N[n]$ is zero-mean white noise with a standard deviation of 0.005 fraction bound. **c)** The response of the receptors was sampled at $f_s = 1/15 \text{ Hz}$ and the noisy samples were digitally low-pass filtered with $f_T = 1/400 \text{ Hz}$. The data was then used to reconstruct the original target concentration either using the pre-equilibrium TEA (solid lines) or conventional inverse-Langmuir (dashed lines) for the receptor with $K_D = 1 \text{ nM}$ (red) or 10 nM (orange). **d)** We calculated the error in the target concentration after repeating the simulation with receptors with $k_{on} = 10^6 \text{ s}^{-1}\text{M}^{-1}$ but

a range of K_D s. This is shown in terms of SNR for both the pre-equilibrium TEA (solid line) and the simple inverse-Langmuir (dashed line). Vertical red and orange lines correspond to the simulations in **b** and **c**.

7. The reviewer suggested a figure that plots imprecision and inaccuracy of the concentration measurements as a function of K_D , both with and without TEA applied: “The manuscript compares sensor performance for sensors with different affinity parameters. I think a figure should be added that plots the concentration measurement imprecision and inaccuracy of the biosensor as a function of K_D , showing curves with as well as without TEA applied. This will help readers to understand the advantages of different approaches (e.g. change K_D versus apply signal processing strategy).”

We agree with the reviewer and believe that the new Figure 8 shown above (especially panel c) addresses this comment.

8. The reviewer asked for a more thorough assessment of the impact of applying the TEA: “The impact of applying the TEA is insufficiently clear to me. Will the method lead to fundamentally new levels of sensing functionality that are widely applicable, or are the improvements incremental and suited for a limited biosensor scope? This requires quantitative extrapolations based on theoretical as well as experimental results.”

We believe pre-equilibrium analysis should be applicable to any biosensor that quantifies target concentrations via ensemble-based measurements of large numbers of receptors, particularly when the kinetics of the receptors are mismatched to the physiological concentration dynamics. Additionally, the TEA described in this work is suitable for biosensors that are well-characterized by the Langmuir isotherm and operate through a two-state binding mechanism. Affinity reagents such as antibodies, recombinant antibody fragments, peptide/small molecule binders, and certain aptamers all operate in this fashion or are well-approximated by this model. However, affinity reagents with more complex binding schemes could be employed in pre-equilibrium context as well if the differential equation describing their operation is well-characterized. The TEA would then simply take the form of the re-arranged differential equation.

Fundamentally, we believe that current equilibrium-based sensors that only leverage the Langmuir and inverse-Langmuir are ‘leaving information on the table’ by not considering the equilibration dynamics of the receptor. Valuable information is contained in these transient signals that should not be ignored, particularly since it can be leveraged to decouple the inherent relationship between sensitivity and kinetics.

Other Points, Reviewer 1:

- **The reviewer felt the section between ‘Principles and limitations of equilibrium sensing’ and ‘Frequency response of the receptor’ seems superfluous.**

We appreciate the reviewer's expertise in this field. However, we believe that these sections should be a part of the main text as they provide important context that may help experimentalist biosensor designers develop the necessary background to be able to understand pre-equilibrium techniques and how to apply them to their devices.

- **The reviewer asked, “The conclusion states that the optimal K_D is larger than T_0 . This seems according to expectation, because a sensor with $K_D < T_0$ approaches saturation and becomes insensitive to concentration fluctuations. Is this viewpoint correct?”**

Conventionally, the dynamic range of a Langmuir-based biosensor is defined as the concentration regime bounded by $0.11 \times K_D$ and $9 \times K_D$ where the response is approximately linear. Outside of this regime, the equilibrium response of the sensor is very weak *in both directions*. For this reason, sensor designers typically require a receptor with a K_D that is very well-matched to the endogenous concentrations observed. This work reveals that when pre-equilibrium is used and noise is considered, the location of the optimal K_D is more nuanced and depends on the kinetics of the receptor, among other things.

When the sensor is at equilibrium and the inverse-Langmuir is applied, a fixed amount of fraction bound noise results in the least amount of concentration uncertainty if $K_D \approx T_0$, such that the fraction bound noise occurs at the steepest part of the isotherm and results in the smallest error in concentration. For a given k_{on} , changing k_{off} such that $K_D > T_0$ or $K_D < T_0$ results in more noise. This is no longer true if the sensor is employed in pre-equilibrium and the TEA is used instead. As discussed in the text, the TEA applies frequency compensation, amplifying high frequencies in the fraction bound signal, to estimate the target concentration before the sensor reaches equilibrium. This results in amplification of noise as well. The amount of amplification depends on the kinetics of the receptor. Thus, for a given k_{on} , the optimal choice of k_{off} needs to consider the TEA compensation in addition to a well-matched K_D . Our analysis shows that in many cases, it is advantageous to trade a mismatched K_D for faster k_{off} , and thus less noise amplification. As shown in Eq. 13, the interplay of these factors is complicated by other parameters such as f_T and f_S as well.

Reviewer 2:

We thank the reviewer for the favorable view of our work and for the thoughtful comments and suggestions. We address these below.

- **The reviewer stated, “It may be helpful to add a sentence on how the timescale for equilibration depends on k_{on} and k_{off} to close the argument being made (it is stated on line 114 for the condition where the number of receptors is small compared to number of targets). A brief discussion of mass transport effects that play a role when the receptors bind and deplete the targets would also be helpful, either on page 3 or on page 5.”**

We agree with the reviewer's suggestion, and have added the following sentence (in red) to the introduction:

The on-rate, k_{on} , is subject to a fundamental upper limit that is determined by the physical and structural properties of the receptor and target; values range between $10^6 - 10^7 \text{ s}^{-1}\text{M}^{-1}$ for typical proteins, with a theoretical upper limit of $\sim 10^8 \text{ s}^{-1}\text{M}^{-1}$ (Refs 2–4). Consequently, differences in affinity are largely due to differences in k_{off} , where higher receptor affinity requires a slower off-rate². **Because the rate of equilibration is given by $k_{eq} = k_{on}[T] + k_{off}$, this results in long equilibration times.** Most molecular biosensors are ‘endpoint’ sensors, designed to measure target concentrations in a sample of biofluid at a single point in time. In this context, slow equilibration kinetics can be remedied by allowing more time for the sensor to reach equilibrium, and thus are not a concern beyond introducing greater delay in generating a final sensor readout.

Regarding mass transport effects and target depletion, we now discuss this and other topics at length in the newly added Appendix 1.

- **The reviewer queried, “is the assumption that the number of target molecules bound is sufficiently high that the noise in y is negligible typically a valid one? Can briefly state the assumption and under what conditions it is valid. It will be helpful to provide an expression for the validity of the assumption that the target concentration is unaffected by the receptors. The assumption should be OK when the timescale for equilibration ($1/k_{eq}$) is larger than the timescale for diffusion across a volume given by the number of targets bound to the receptors at equilibrium divided by the target concentration.”**

We agree with the reviewer, and address this in the newly added Appendix 1 as well.

- **The reviewer requested, “it would be helpful for the sake of clarity to add an equation stating that the measured signal is given by $w[n] + \text{noise}$, and that $w[n]$ depends on y by some constant linear scaling factor.”**

We agree and have applied this change to the text:

In this new model, the sensor samples these continuous-time signals with sampling frequency f_s and generates a discrete-time sequence of measurements, $y[n]$, where the integer variable n replaces the continuous variable t . **Measurement noise, $N[n]$, is an additive term introduced by the sensor, leading to the noisy sensor output $w[n] = y[n] + N[n]$.** The revised TEA is the re-arranged law of mass action (Eq. **Error! Reference source not found.**) applied to $w[n]$, which generates a discrete-time sequence of target estimates, $E[n]$. In the absence of noise, this system would accurately track the target concentration regardless of receptor kinetics – we used this model to understand how measurement noise impairs accurate target estimation.

- **The reviewer asked us to briefly state the key assumptions on page 12, line 230: “need to briefly state the key assumptions in the analysis here.”**

We have added the following text:

This analysis assumes that the time-domain response of the system to the periodic sinusoidally-varying input is itself periodic and can be expressed as a Fourier series. Additionally, we assume that the magnitude of the oscillation of the fraction bound is small compared to the mean value. We then run simulations to evaluate the accuracy of the model when this is not the case.

- **The reviewer stated, “It will be helpful to clarify somewhere that the same sensor can be functionalized with different receptors. This will justify the assumption that the sensor noise is independent of the receptor properties. A sketch in figure 1 or a figure before figure 1 may be helpful.”**

We agree with the reviewer and have added the following portions of text:

Furthermore, later in this work we show that, while the pre-equilibrium method could be applied to sensors functionalized with any receptor, measurement precision can be maximized by optimizing the kinetic properties of the receptor employed by the sensor.

The dominant components of this noise depend on the nature of the detection modality, but we assume here that $N[n]$ encompasses all relevant sources of random uncertainty. We also make the simplifying assumption that $N[n]$ is white gaussian noise with equal power (given by $N_0/2$ with units of (fraction bound)²/Hz) at all frequencies, and that this noise is independent of the receptor used.

- **The reviewer requested a glossary of symbols.**

We agree and have added such a glossary as Appendix 6.

Reviewer 3:

- **The reviewer asked if this pre-equilibrium process is energetically favorable: “Since authors describe this work as to achieve sensing ability of the device before the equilibrium is reached, that is, in pre-equilibrium state (non-equilibrium process), therefore, I would like to know is this process thermodynamically favorable? What about the Gibbs free energy change for this pre-equilibrium process?”**

We would like to clarify that the method proposed in this work does not involve changing the operation of the molecular receptor itself. As such, the ligand-receptor binding reaction is

thermodynamically favorable, with a negative Gibbs free energy, and will eventually reach equilibrium. The pre-equilibrium method described in this paper simply involves observing the rate-of-change of bound receptors as the reaction tends towards equilibrium. This allows us to estimate the target concentration *before* equilibrium is reached.

- **The reviewer asked how we can practically enhance receptor kinetics: “A receptor with fast kinetics will respond equally well to low- and high-frequency content. How can we enhance the kinetic of the receptor for real application following of this strategy?”**

Depending on the nature of the receptor (*e.g.*, antibody, peptide, aptamer, etc.), the kinetics of the receptor could be improved by engineering or screening mutations at one or more locations in the structure to achieve a faster off-rate.

However, the intention of this work is to consider cases where the kinetics of the receptor are insufficient to successfully track concentration changes. The main finding of the work is that a mathematical method of compensating for slow receptor kinetics exists, and that it can be leveraged to reconstruct rapidly changing target concentrations. As shown in the text, when this is done, designers must pay careful attention to noise levels in the detector to ensure that they are not excessively large.

REVIEWER COMMENTS

Reviewer #1 (Remarks to the Author):

I would like to thank Maganzini et al. for revising their manuscript.

The most important point was to add experimental data. In fig. 3, the authors report standard BLI sensorgrams, where first analyte is added (20 nM) followed by a blank measurement (0 nM). However, the data, data analysis, and discussion are not convincing:

1. In fig. 3, the authors decided to remove certain measurement data (between 0 and 30 s, and between 270 and 320 s). Measurement data should not be removed, because a continuous sensor should operate continuously. All data should be shown, analyzed, and discussed.
2. The authors have data available for a large range of concentrations (fig. S1). All data should be analyzed using I-L and TEA. The data should be shown as time traces (as in fig. 3) and in concentration correlation graphs. A correlation graph has on the x-axis the applied analyte concentrations (I would suggest: datapoints at $t=100$, $t=200$, $t=400$, and $t=600$) and on the y-axis the concentrations reported by the sensor, analyzed by I-L as well as by TEA.
3. The authors state that non-specific binding plays a role. Non-specific binding is very common in biosensors and is indeed present in the sensors of fig. 3. Therefore, non-specific binding should be added to the model, for both analysis methods (I-L and TEA).
4. Accuracy is a key property of a biosensor that reports concentrations. The data in fig. 3 show large discrepancies between the applied analyte concentrations (20 nM between 0s and 300s, and 0 nM between 300s and 600s) and the concentrations reported by the sensor, whichever data analysis method is used (I-L or TEA). This means that the reported sensor and analysis method are highly inaccurate. This should be quantitatively studied and solved.
5. Finally, to validate the concept of continuous measurements, at least one experiment should be done with a longer series of concentrations applied, using at least 6 concentrations applied in series on the same sensors, e.g. 20 nM, 0 nM, 20 nM, 0 nM, 20 nM, 0 nM.

Reviewer #2 (Remarks to the Author):

The authors have satisfactorily addressed the reviewer concerns. The reviewer has one comment before publication (no further review required) - What are the uncertainties in the fitted k_{on} and k_{off} values for the newly added experimental data? These should be provided. What is the level of residual non-specific binding, and how is it quantified? Presumably, when an exponential decay curve is fitted to the data with an offset as a free parameter, the residual binding is obtained as the offset?

Response Letter:

Reviewer 1:

We thank the reviewer for the thoughtful examination of our work. We have addressed his/her comments on our revised manuscript below.

1. The reviewer requested that all experimental data should be included in the experimental figure 3: “In fig. 3, the authors decided to remove certain measurement data (between 0 and 30 s, and between 270 and 320 s). Measurement data should not be removed, because a continuous sensor should operate continuously. All data should be shown, analyzed, and discussed.”

As requested by the reviewer, we have amended Figure 3 to include all data. We believe the figure clearly conveys the performance improvements conferred by the proposed TEA algorithm relative to the conventional inverse-Langmuir method.

Figure 3: Experimental proof of principle. a) Normalized biolayer interferometry (BLI) binding data for two monoclonal antibodies, mAb1 (red) and mAb11 (orange), exposed to 20 nM TNF α from 0–300 s and to 0 nM TNF α from 300–600 s. Black lines represent digitally low-pass filtered data. The target concentration was estimated using the pre-equilibrium TEA (solid lines) as well as the conventional inverse-Langmuir (dashed lines) for b) mAb1 and c) mAb11 data. Solid black lines represent the inverse-Langmuir target estimates based on the fraction of bound receptors at equilibrium.

2. The reviewer requested that all BLI data used for antibody characterization be included in the inverse-Langmuir vs. TEA comparison. Additionally, the reviewer requested concentration-correlation graphs: “The authors have data available for a large range of concentrations (fig. S1). All data should be analyzed using I-L and TEA. The data should be shown as time traces (as in fig. 3) and in concentration correlation graphs. A correlation graph has on the x-axis the applied analyte concentrations (I would suggest: datapoints at $t=100$, $t=200$, $t=400$, and $t=600$) and on the y-axis the concentrations reported by the sensor, analyzed by I-L as well as by TEA.”

As requested, we have performed inverse-Langmuir vs TEA comparison for all TNF α concentrations (**Fig. S2a, b**) and have added these to Appendix 2. These figures also include concentration-correlation graphs for the 15 s, 100 s and 200 s time-points (**Fig. S2c**). For the 400 s and 600 s time-points in blank solution, we have plotted the estimated target concentration of both the traditional inverse-Langmuir and the TEA as a function of the previous target concentration to which that antibody was exposed (**Fig. S2d**).

These data confirm our previous findings: the proposed TEA algorithm instantaneously predicts concentrations of target that are very close to the values that would only be predicted by the inverse-Langmuir after very long equilibration times. The TEA correctly estimates all concentrations (< 15% error across all concentrations) in less than 15 s, whereas the conventional inverse-Langmuir method has comparatively poor accuracy even after 200 s of exposure to solution. We observe similar outcomes across a range of target concentrations, where the TEA algorithm generates target estimates within less than a minute that match those that can only be obtained after very long equilibration times with the conventional I-L method.

Figure S2: Additional proof-of-principle data. **a)** Normalized biolayer interferometry (BLI) binding data using mAb11. From 0–300 s, the probe was exposed to different TNF α concentrations; from 300–600 s, the probe was dipped in blank solution. Black lines represent digitally low-pass filtered data. **b)** The filtered data was used to estimate the target concentration using the pre-equilibrium TEA (solid lines) as well as the conventional inverse-Langmuir (dashed lines). Solid black lines represent the inverse-Langmuir target estimates based on the fraction of bound receptors at equilibrium. Dotted black lines represent the true target concentration in solution. **c)** The target estimates at $t = 15$ s, 100 s and 200 s for both TEA and inverse Langmuir were extracted and plotted against the actual target concentrations in solution. Dotted black line corresponds to equality between predicted and actual concentrations. **d)** TEA and inverse-Langmuir target estimates at $t = 400$ s and 600 s, which correspond to when all BLI probes were in blank solution, were extracted and plotted against the concentration to which the probes were exposed prior to the blank sample. Black diamonds show the predicted concentration if the sensor reached equilibrium.

3. The reviewer expressed concern regarding non-specific binding in the data presented and requested that non-specific binding be added to the model: “The authors state that non-specific binding plays a role. Non-specific binding is very common in biosensors and is indeed present in the sensors of fig. 3. Therefore, non-specific binding should be added to the model, for both analysis methods (I-L and TEA).”.

We respectfully disagree with the reviewer’s suggestion that non-specific binding should be added to the model. Non-specific binding is dictated by a wide range of parameters, including sample type, device geometry, device passivation, and receptor type, and is thus an inherently unpredictable phenomenon. As such, accounting for non-specific binding in a continuous biosensor where target concentrations are constantly changing remains an unsolved problem. Instead, biosensor developers employ engineering solutions that reduce non-specific binding (e.g., through surface passivation or protection with a semi-permeable membrane) and/or make the binding signal more specific (e.g., using an electrochemical or fluorescent reporter). This remains an active area of research, and the detailed characterization of non-specific binding in BLI sensors is far beyond the scope of this work.

Instead, our focus is on providing a framework for extracting accurate information about fast-changing biomarkers from sensors with slow receptors. The motivation for including these BLI data is to experimentally demonstrate the advantage of implementing the pre-equilibrium TEA over the existing inverse-Langmuir estimation method, and thereby removing the requirement for sensor equilibration. As such, we believe it is justified to omit a more complex discussion of non-specific binding, beyond demonstrating that our proposed pre-equilibrium method does not exacerbate its impact on sensing accuracy relative to the conventional inverse-Langmuir method. As discussed in our response to the reviewer’s comment #5 in the previous response letter, TEA does not amplify errors caused by non-specific binding, which instead appear as low-frequency offsets or drift. In fact, these drift and offset effects will impair a pre-equilibrium sensor exactly the same as existing equilibrium inverse-Langmuir sensors. Accordingly, we do not believe that

readers would derive additional value from further efforts to characterize non-specific binding in this particular study.

4. The reviewer expressed raised questions about discrepancies in the reported target estimates from both data analysis methods, and asked us to investigate this further: “Accuracy is a key property of a biosensor that reports concentrations. The data in Fig. 3 show large discrepancies between the applied analyte concentrations (20 nM between 0s and 300s, and 0 nM between 300s and 600s) and the concentrations reported by the sensor, whichever data analysis method is used (I-L or TEA). This means that the reported sensor and analysis method are highly inaccurate. This should be quantitatively studied and solved.”

We thank the reviewer for raising this question. We would like to remind the reviewer that BLI was designed to measure binding properties at single time-points, and not for continuous monitoring - this is the main source of error contributing to the observed measurement inaccuracies.

That being said, the priority of this work was to introduce a novel algorithmic approach for improving the sensitivity and accuracy that can be achieved *with existing sensor platforms*. Towards this goal, we focused on demonstrating how TEA can produce more accurate results than the “gold-standard” inverse-Langmuir regardless of the input data. As suggested by the reviewer, we employed standard BLI techniques to generate a representative dataset for continuous molecular binding measurements. Indeed, we believe that the limitations of BLI probe are important here, as they prove that the TEA algorithm does not exacerbate the impact of these non-idealities on target estimation.

5. The reviewer requested a longer continuous sensing experiment in which concentrations are cycled repeatedly: “Finally, to validate the concept of continuous measurements, at least one experiment should be done with a longer series of concentrations applied, using at least 6 concentrations applied in series on the same sensors, e.g. 20 nM, 0 nM, 20 nM, 0 nM, 20 nM, 0 nM.”.

We thank the reviewer for this comment. However, we would like to kindly remind the reviewer that the main point of the work is not to convert BLI into a continuous sensor. Rather, our focus is on providing a novel theoretical framework that exceeds the performance of conventional inverse-Langmuir approach for extracting concentration information using sensors with slow receptors. We have already shown that our algorithm enables faster, and more accurate quantification compared to the inverse-Langmuir approach for both increasing and decreasing concentrations.

In practice, performing multiple cycles requested by the reviewer would be a challenge for the BLI sensor - not our algorithm. This is because BLI was not designed to be a continuous sensor – it uses refractive index changes to measure analyte concentrations and thus it suffers from challenges of avidity and non-specific binding. As such, we believe that additional cycles of BLI would require considerable effort. In essence, we need to convert the BLI into a continuous sensor –

clearly, this is far beyond the scope of this work and does not add to the effectiveness of our TEA algorithm.

Reviewer 2:

Reviewer 2 did not request further review and suggested accepting the manuscript - we thank the reviewer for his/her support. But the reviewer provided the following comments:

“What are the uncertainties in the fitted k_{on} and k_{off} values for the newly added experimental data?” These should be provided.

As suggested by the reviewer, we have added the uncertainties in kinetic rates to the manuscript.

“What is the level of residual non-specific binding, and how is it quantified? Presumably, when an exponential decay curve is fitted to the data with an offset as a free parameter, the residual binding is obtained as the offset?”

As the reviewer notes, when a biosensor is exposed to an initial concentration followed by a sudden switch to blank solution, the receptor equilibration will follow an exponential decay. As such, by including an offset as a free parameter, one can quantify non-specific binding on the sensor at a given time-point. However, in this work, we are analyzing scenarios in which one is determining a target concentration that is constantly changing in time. As such, the proposed method cannot be used to quantify the level of non-specific binding in this context. Indeed, accounting for non-specific binding in a continuous biosensor remains an unsolved problem, and researchers have therefore engineered various solutions that reduce non-specific binding (*e.g.*, through surface passivation) and/or make the binding signal more specific (*e.g.*, using an electrochemical reporter).

REVIEWERS' COMMENTS

Reviewer #1 (Remarks to the Author):

MAIN REMARKS

I thank the authors for adapting the manuscript and writing a rebuttal.

The main question now is: what is needed to make this work a high-impact work?

The present paper describes a new and elegant data analysis method (eq. 3) and supports it with simulations. In my opinion, this work can become a high-impact paper when the new data analysis method is sufficiently validated with experiments. The experiments are not difficult. What is required is willingness to test the theory with experiments and thereafter extend the theoretical framework so that it achieves agreement with the experiments. After this, the paper will contain a tested concept that can be used by many scientists.

1. The old approach is the inverse equilibrium Langmuir analysis, which contains one free parameter (K_d) and fraction y . The new approach is equation 3, which has two free parameters (k_{on} , k_{off}), fraction y , and its time derivative (dy/dt). The fact that the new approach has more free parameters and a time derivative, means that noise (due to the derivative) and systematic errors (due to more free parameters) need to be treated.

2. Pre-equilibrium biosensing, i.e. the fact that a target concentration can be derived from the slope of a binding curve, is well-known in literature, see e.g. Liu, Sci.Rep. 2015 Surface Plasmon Resonance Biosensor Based on Smart Phone Platforms: "by monitoring the relative intensity and slopes in an experiment, we can determine the concentration and obtain binding-kinetics information". One may wonder: as it has been known for a long time that the slope of a binding curve can reveal the target concentration, why do assays (in in-vitro diagnostics and in research) derive the concentration from a measured equilibrium level, i.e. an inverse equilibrium Langmuir analysis, rather than from the slope? The reason is that the equilibrium has lower uncertainty (only one free parameter, namely K_d); the calibration is simpler and deviations in concentration readings (systematic & noise) are smaller than in slope-based assays. The message is: when introducing a new analysis method, it is essential to explain how small deviations can be achieved between applied target concentrations and concentrations reported by the analysis method. The authors need to demonstrate to the readers not just a single equation, but explain a data analysis solution that achieves small deviations in an experiment.

3. The authors stress how the new data analysis method performs between 0 and 200 s in fig. 3. However, that phase is a standard biosensing phase, as reported in earlier literature (see above). The interesting continuous biosensing occurs between 200 and 400 s, when the target concentration transitions from a high to a low level (20 nM to 0 nM); that's the phase where the performance of the new data analysis method needs to be judged. Here we see that the deviations (deviation = difference between applied target concentrations and the concentrations reported by the analysis method) are large, in and after the transition. These deviations should be discussed in detail. This shows that the proposed data analysis method is not yet complete and not yet broadly applicable.

4. The fact that the TNFalpha assay in fig. 3 has strong non-specific binding is not a valid excuse to halt the experimental validation. All sensors have degrees of non-specific binding. The authors can characterize the non-specific binding behavior of their assay and include it in the model; alternatively, they can use an assay that has low non-specific binding, see many examples in the literature.

5. The present manuscript assumes that systematic errors are the same for the old and the new analysis method. This assumption is incorrect, as explained in points 1 and 2. Another difference: the new method can give negative concentration values in phases where the slope dy/dt is negative. The old data analysis method does not give negative concentration values. In fig. 3b and 3c negative concentrations are not seen due to the erroneous upward shift, but negative concentrations will become visible when non-specific binding is absent or corrected for in the BLI experiment. This stresses the importance of studying systematic deviations and not only noise. The new analysis method will be broadly interesting and applicable when it is clearly demonstrated how systematic errors can be dealt with and what the results will be.

6. A data analysis method for continuous biosensing should be able to operate with increasing concentrations, decreasing concentrations, and transitions between those two. Therefore I think at least one experiment should be done with a series of concentrations, e.g. 20 nM, 0 nM, 20 nM, 0 nM, 20 nM, 0 nM.

OTHER REMARKS

1. The authors mention that in their opinion, BLI is not a continuous biosensing method, which I think is a curious statement. SPR, QCM and BLI give immediate and continuous responses to changing analyte concentrations. SPR, QCM and BLI are widely available and easy to work with. These are very good platforms to validate a newly proposed data analysis method to convert measured binding signals into target concentrations.

2. Fig. 3b, 3c, and S2b: The thick black curves are misleading. These curves suggest that TEA gives the right values of target concentration. However, the thick black curves are systematically wrong, because the real concentrations are 20 nM at start and 0 nM after 300 s, as shown in fig. 3a. In panels 3b, 3c, and S2b, the real concentrations should be emphasized, not the erroneous target concentrations.

Response to reviewer's comments:

1. “The old approach is the inverse equilibrium Langmuir analysis, which contains one free parameter (K_D) and fraction y . The new approach is equation 3, which has two free parameters (k_{on} , k_{off}), fraction y , and its time derivative (dy/dt). The fact that the new approach has more free parameters and a time derivative, means that noise (due to the derivative) and systematic errors (due to more free parameters) need to be treated.”

We agree with the reviewer about the importance of noise when using a time derivative and this is why we dedicated a considerable section analyzing it mathematically in our manuscript. The frequency-domain equations reveal that the added noise in our pre-equilibrium approach can be minimized by leveraging our data analysis method described in our manuscript. Importantly, we have explicitly shown that the pre-equilibrium TEA does not amplify error caused by offsets, systematic errors, and other low-frequency drifts. This is shown analytically in equations 10 and 11 and through numerical simulation in Figure 6c in the manuscript, where we show that the TEA and inverse-Langmuir are equally affected by low frequency signals such as offsets and drifts. Furthermore, we also show this experimentally in the BLI data, where the error caused by non-specific binding is identical for both the inverse-Langmuir and the TEA. Indeed, despite the increased noise, our approach achieves superior accuracy to the conventional inverse-Langmuir approach.

Finally, we respectfully disagree with the reviewer's view regarding the use of K_D versus k_{on} and k_{off} to characterize receptor binding properties. For many affinity reagents and particularly for antibodies—the focus of this manuscript—the gold-standard method for determining K_D is directly based on measurements of k_{on} and k_{off} from methods such as BLI or SPR, where $K_D = k_{off}/k_{on}$. Consequently, any uncertainty in kinetics is directly coupled to the uncertainty in K_D .

2. “Pre-equilibrium biosensing, i.e. the fact that a target concentration can be derived from the slope of a binding curve, is well-known in literature, see e.g. Liu, Sci.Rep. 2015 Surface Plasmon Resonance Biosensor Based on Smart Phone Platforms: “by monitoring the relative intensity and slopes in an experiment, we can determine the concentration and obtain binding-kinetics information”. One may wonder: as it has been known for a long time that the slope of a binding curve can reveal the target concentration, why do assays (in in-vitro diagnostics and in research) derive the concentration from a measured equilibrium level, i.e. an inverse equilibrium Langmuir analysis, rather than from the slope? The reason is that the equilibrium has lower uncertainty (only one free parameter, namely K_D); the calibration is simpler and deviations in concentration readings (systematic & noise) are smaller than in slope-based assays. The message is: when introducing a new analysis method, it is essential to explain how small deviations can be achieved between applied target concentrations and concentrations reported by the analysis method. The authors need to demonstrate to the readers not just a single equation, but explain a data analysis solution that achieves small deviations in an experiment.”

The law of mass action is indeed well-known to the scientific community, but we disagree with the reviewer's interpretation of why it has not been widely used in the realm of continuous biosensing. Calibration is not the issue—in fact, our experimental data show that the parameters

required for pre-equilibrium operation can be easily obtained with a simple 30-minute BLI experiment and standard kinetic analysis techniques. Rather, the limiting factor has been the impact of detector noise. As noted above and in our manuscript, due to the time derivative the impact of noise on the TEA is larger than on the inverse-Langmuir and the relationship between system parameters and noise is notably more complex than with the inverse-Langmuir.

This was our motivation for pursuing this study, and we dispute the reviewer's reduction of the work to "just a single equation". Indeed, we have provided a full frequency-domain characterization of the behavior of this equation and have used this characterization to inform methods of optimizing performance of this equation on both the receptor and the data-processing side. And as shown by our experimental data, implementation of this approach leads to vast improvements in biosensing compared to the inverse-Langmuir, which we believe is clearly an important and notable finding that supports the utility and value of our approach.

3. "The authors stress how the new data analysis method performs between 0 and 200 s in fig. 3. However, that phase is a standard biosensing phase, as reported in earlier literature (see above). The interesting continuous biosensing occurs between 200 and 400 s, when the target concentration transitions from a high to a low level (20 nM to 0 nM); that's the phase where the performance of the new data analysis method needs to be judged. Here we see that the deviations (deviation = difference between applied target concentrations and the concentrations reported by the analysis method) are large, in and after the transition. These deviations should be discussed in detail. This shows that the proposed data analysis method is not yet complete and not yet broadly applicable."

We respectfully disagree with the reviewer. Nothing in our analysis suggests that the directionality of change should impact the quality of estimation. As discussed in the manuscript, the behavior of the pre-equilibrium algorithm is analogous to that of an inverse low-pass frequency filter: it modifies the signal based on its frequency content. This is consequential in the context of noise, because certain frequencies of noise will be amplified in the target estimate. However, the polarity or directionality of the change of target signal has no bearing on this behavior, and we would expect the noise propagation to behave as described by the equations for any arbitrary sequence or "shape" of target concentration changes.

To this end, we would call the reviewer's attention to Figure S2, which was added in the previous round of revision and is copied below. The reviewer had specifically asked for concentration correlation graphs to compare expected concentrations with achieved estimates at 100, 200, 400, and 600 s. We found that at every time-point and interval between time-points (*e.g.* between 250–350 s), our method performs better than the existing analysis method.

Finally, the deviations cited by the reviewer have in fact already been discussed at length, both in previous response letters and the manuscript itself. These deviations are due to BLI sensor limitations, not limitations in the analysis method. Since our focus here is on describing and validating a novel algorithmic approach for data analysis, our primary concern here was to demonstrate improved performance relative to the existing algorithm, and we believe the mathematical analyses, numerical simulations, and experimental data that we have provided all strongly support this claim.

Figure S2: Additional proof-of-principle data. **a)** Normalized biolayer interferometry (BLI) binding data using mAb11. From 0–300 s the probe was exposed to different TNF α concentrations; from 300–600 s, the probe was dipped in blank solution. Black lines represent digitally low-pass filtered data. **b)** The filtered data was used to estimate the target concentration using the pre-equilibrium TEA (solid lines) as well as the conventional inverse-Langmuir (dashed lines). Solid black lines represent the inverse-Langmuir target estimates based on the fraction of bound receptors at equilibrium. Dotted black lines represent the true target concentration in solution. **c)** The target estimates at $t = 15$ s, 100 s and 200 s for both TEA and inverse Langmuir were extracted and plotted against the actual target concentrations in solution. Dotted black line corresponds to equality between predicted and actual concentrations. **d)** TEA and inverse-Langmuir target estimates at $t = 400$ s and 600 s, which correspond to when all BLI probes were in blank solution, were extracted and plotted against the concentration to which the probes were exposed prior to the blank sample. Black diamonds show the predicted concentration if the sensor reached equilibrium.

4. “The fact that the TNFalpha assay in fig. 3 has strong non-specific binding is not a valid excuse to halt the experimental validation. All sensors have degrees of non-specific binding. The authors can characterize the non-specific binding behavior of their assay and include it in the model; alternatively, they can use an assay that has low non-specific binding, see many examples in the literature.”

We agree with the reviewer that all sensors have degrees of non-specific binding. As we described in our previous response letter, non-specific binding is dictated by a wide range of parameters, including sample type, device geometry, device passivation, and receptor type, and is thus an inherently unpredictable phenomenon. As such, there is no general way of including this in the model that would be valid for a continuous biosensor where target concentrations are constantly changing along an unknown trajectory. Finding a solution to this still-unsolved problem thus falls well beyond the scope of this work.

5. “The present manuscript assumes that systematic errors are the same for the old and the new analysis method. This assumption is incorrect, as explained in points 1 and 2. Another difference: the new method can give negative concentration values in phases where the slope dy/dt is negative. The old data analysis method does not give negative concentration values. In fig. 3b and 3c negative concentrations are not seen due to the erroneous upward shift, but negative concentrations will become visible when non-specific binding is absent or corrected for in the BLI experiment. This stresses the importance of studying systematic deviations and not only noise. The new analysis method will be broadly interesting and applicable when it is clearly demonstrated how systematic errors can be dealt with and what the results will be.”

We have addressed the reviewer’s concerns about systematic errors and baseline shifts above. However, the reviewer is incorrect that the inverse-Langmuir method cannot result in negative concentrations in the presence of noise. In the absence or near-absence of target, random detector noise can lead to observed signals that are below the signal level predicted by the inverse-Langmuir method. In other words, these signals look like a negative concentration. In practice, however, these events are easily addressed in existing biosensor and assay implementations by reporting a value of 0 M, which is the closest physically reasonable concentration estimate. Thus, this “negative concentration” phenomenon is not a unique to our proposed method, nor is it a challenging problem to solve for standard sensor implementations. And as we have explained above, the impact of detector noise on the pre-equilibrium method is one of the main foci of this paper, which is why we have carefully characterized how noise propagates through this system as well as strategies to mitigate its impact, whereas specific details of sensor implementation are not.

6. A data analysis method for continuous biosensing should be able to operate with increasing concentrations, decreasing concentrations, and transitions between those two. Therefore I think at least one experiment should be done with a series of concentrations, e.g. 20 nM, 0 nM, 20 nM, 0 nM, 20 nM, 0 nM.

We agree with the reviewer’s general premise, and we have already demonstrated all of these operating conditions in the proof-of-concept data shown in Figure S2. These data show that the mathematical operations we propose in this work are valid in regimes with increasing

concentration, decreasing concentration, or transition between the two. However, the additional experiment proposed by the reviewer would not be a productive or meaningful extension of this work. Such concentration cycling would not affect the performance of our analytical framework, but it would be a challenge for the BLI sensor, which is not intended for use as a continuous biosensor. That is because this method was designed to extract kinetic constants using simple exponential fits in experiments where target concentrations are known and change at predetermined intervals, and is optimized only for measuring single, stepwise concentration changes rather than ongoing fluctuations within a given sample.

7. “The authors mention that in their opinion, BLI is not a continuous biosensing method, which I think is a curious statement. SPR, QCM and BLI give immediate and continuous responses to changing analyte concentrations. SPR, QCM and BLI are widely available and easy to work with. These are very good platforms to validate a newly proposed data analysis method to convert measured binding signals into target concentrations.”

As noted in our response to comment number 6, BLI—and likewise, methods like SPR and QCM—can indeed generate binding data in a continuous fashion, *but are not capable of continuously tracking fluctuations in target concentration in a given sample*. Over the course of a BLI experiment, the BLI probe is moved between discrete wells containing different concentrations of target protein. Thus, while BLI can provide time-resolved measurements in this fashion, it is not a true continuous biosensor as defined in the context of our work.

8. “Fig. 3b, 3c, and S2b: The thick black curves are misleading. These curves suggest that TEA gives the right values of target concentration. However, the thick black curves are systematically wrong, because the real concentrations are 20 nM at start and 0 nM after 300 s, as shown in fig. 3a. In panels 3b, 3c, and S2b, the real concentrations should be emphasized, not the erroneous target concentrations.”

These black curves are included because they correspond to the fair comparison between the performance of our proposed pre-equilibrium algorithm and the conventional inverse-Langmuir estimates. We want to emphasize these values, because they show that pre-equilibrium immediately produces the same estimate that the existing algorithm would have produced only after a long period of incubation in order to reach equilibrium. We would also like to emphasize that our intent is not to minimize the discrepancy between the solid black lines and the true target concentrations, and we have included text in the manuscript to describe this discrepancy and clearly explain and discuss the source of it. This allows readers to clearly observe how the presence of non-idealities such as non-specific binding affects the existing algorithm and the novel algorithm equally.